 # Swordsman: Entropy-Driven Adaptive Block Partition for Efficient Diffusion Language Models

**Yu Zhang** [1]  **Xinchen Li** [1]  **Jialei Zhou** [1]  **Hongnan Ma** [2]  **Zhongwei Wan** [3]  **Yiwei Shi** [2]
**Duoqian Miao** [1]  **Qi Zhang** [1]  **Longbing Cao** [4]

## Abstract

Block-wise decoding effectively improves the inference speed and quality in diffusion language models (DLMs) by combining inter-block sequential denoising and intra-block parallel unmasking. However, existing block-wise decoding methods typically partition blocks in a rigid and fixed manner, which inevitably fragments complete semantic or syntactic constituents, leading to suboptimal performance. Inspired by the entropy reduction hypothesis (ERH), we recognize that constituent boundaries offer greater opportunities for uncertainty reduction, which motivates us to employ entropy analysis for identifying constituent boundaries. Therefore, we propose Swordsman, an entropy-driven adaptive block-wise decoding framework for DLMs. Swordsman adaptively partitions blocks by identifying entropy shifts between adjacent tokens to better align with semantic or syntactic constituent boundaries. In addition, Swordsman dynamically adjusts unmasking thresholds conditioned on the real-time unmasking status within a block, further improving both efficiency and stability. As a training-free framework, supported by KV Cache, Swordsman demonstrates state-of-the-art performance across extensive evaluations. Our code is now available.

## 1. Introduction

In recent years, diffusion language models (DLMs) (Austin et al., 2021; He et al., 2023; Gong et al., 2022) have rapidly emerged as a promising alternative to autoregressive language models by breaking the strictly sequential token decoding bottleneck through parallel decoding of multiple tokens at each denoising step. Large-scale DLMs such as LLaDA (Nie et al., 2025; Zhu et al., 2025a;b), Dream (Ye et al., 2025), Mercury (Khanna et al., 2025), and Gemini Diffusion (Google DeepMind, 2025) have demonstrated the feasibility and scalability of the diffusion paradigm in large-scale language modeling. However, limited by diffusion's inherently numerous iterative denoising steps, DLMs currently face a severe challenge: while they possess a high theoretical efficiency ceiling, their actual inference speed and generation quality remain inferior to traditional autoregressive models in practice. This gap motivates the community actively explore better diffusion language decoding methods that balance both speed and quality (Li et al., 2025b; Zhong et al., 2026; Hong et al., 2025; Kou et al., 2026).

Block-wise decoding (Wu et al., 2025b; Arriola et al., 2025) has attracted widespread interest as a simple yet effective solution. It partitions the masked sequences into multiple decoding blocks to enable sequential inter-block denoising coupled with parallel intra-block decoding. Owing to their compatibility with the reuse of preceding blocks' KV cache (Wu et al., 2025a; Wang et al., 2025b; Hu et al., 2025) and various scheduling strategies (Huang et al., 2026; Ben-Hamu et al., 2025; Hong et al., 2025), block-wise decoding methods (Wu et al., 2025b; Ma et al., 2025; Chen et al., 2025) achieve an excellent trade-off between speed and quality under the same computational budget. Nevertheless, the introduction of blocks necessitates boundary partition, and the design of the boundary strategy directly impacts decoding performance. Existing boundary strategies typically adopt the fixed-length block partition, where the inherent rigidity and lack of adaptability often cause block boundaries to fail to align with semantic or syntactic constituents. This misalignment leads to tightly coupled tokens within the same constituent to be fragmented across different blocks, as illustrated in Figure 1 (a), increasing token unmasking difficulty and ultimately degrading performance. More generally, modeling dependencies among output variables has been widely recognized as important for reducing effective output-space complexity in structured prediction problems (Kou et al., 2025; Wu et al., 2026a).

This motivates a constituency-aware boundary strategy for

---

[1]Tongji University [2]University of Bristol [3]The Ohio State University [4]Macquarie University. Correspondence to: Duoqian Miao <dqmiao@tongji.edu.cn>.

*Proceedings of the 43rd International Conference on Machine Learning*, Seoul, South Korea. PMLR 306, 2026. Copyright 2026 by the author(s).

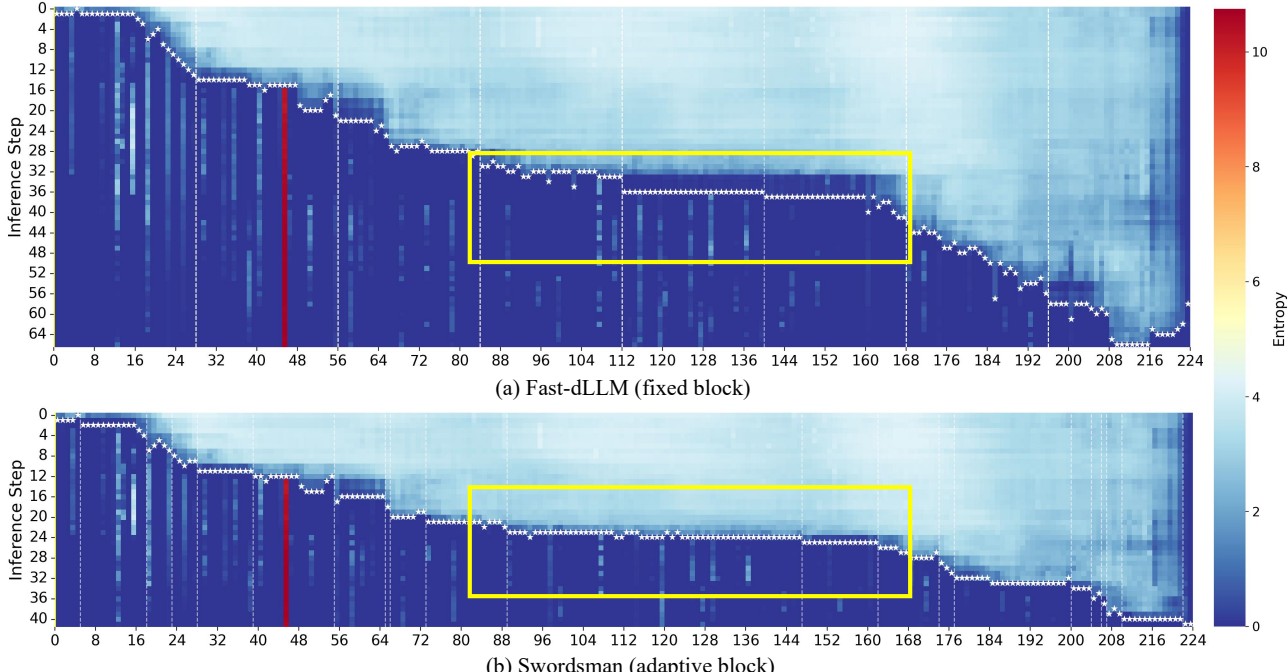

*Figure 1.* Entropy evolution during decoding reveals semantic boundaries through shifts. The y-axis denotes inference steps, the x-axis denotes token positions, and stars mark the tokens unmasked at the current step. Each heatmap value reports the Shannon entropy of the token prediction distribution produced by the final prediction layer. Fast-dLLM (a) ignores entropy variance, applying predetermined fixed-length blocks that frequently fragment coherent constituents or merge semantically distinct ones, thereby degrading generation accuracy. Swordsman (b), however, leverages entropy shifts to adaptively align block boundaries with semantic constituents, partitioning at maximum shift points to achieve precise segmentation that yields better generations.

block-wise decoding. Generally, more important words tend to be more difficult to predict (Levy, 2008). Inspired by the entropy reduction hypothesis (ERH) (Hale, 2006), we interpret the diffusion language generation process as progressively reducing uncertainty about subsequent semantic or syntactic continuations. The hypothesis posits that processing difficulty on a word is positively related to the entropy reduction it induces, i.e., uncertainty about the rest of the sentence after encountering that word. We extend this view to diffusion decoding: constituent boundaries correspond to shifts between syntactic states, and typically exhibit high uncertainty due to the vast freedom in perpetuating constituents. Take the sentence I `DRINK` `ORANGE JUICE` as an example. At the positions of constituents `DRINK` and `ORANGE JUICE`, they admit a huge amount of alternatives for word choice, allowing for the replacement of with any verb or noun, respectively. Conversely, elements within a constituent are tightly coupled and represent continuous expansion of the same syntactic state, tending to exhibit smoother uncertainty evolution. Therefore, when processing words within constituents, the word alternative space (e.g., `()` in `ORANGE ()`) is relatively narrow compared to boundary processing. Consequently, constituent boundaries provide greater potential for uncertainty reduction. While entropy serves as a direct measure of uncertainty. During

diffusion decoding, the model produces token distributions at all positions, allowing us to directly calculate predictive entropy along the sequence. Meanwhile, as shown in Figure 2, we conduct constituency parsing on selected sentences to obtain their constituents and analyze the entropy values at constituent boundaries. The analysis reveals the phenomenon wherein entropy variations generally exhibit large at constituent boundaries. Therefore, we advocate for adaptively partitioning constituent blocks based on the entropy shifts between adjacent tokens.

In light of the above discussion, we propose Swordsman, a training-free block-wise decoding framework for DLMs through entropy-driven adaptive block partition. Specifically, Swordsman calculates the predictive entropy of tokens during unmasking and then identifies constituent boundaries by detecting sharp entropy shifts between adjacent tokens to adaptively partition blocks. Furthermore, considering that adaptive block partition leads to variations in token unmasking scope and difficulty across blocks, Swordsman dynamically adjusts the unmasking thresholds based on the real-time status of parallel unmasking within the block to balance parallelism and reliability within each block.

Extensive experiments and analyses validate Swordsman as a simple yet effective block-wise decoding framework.

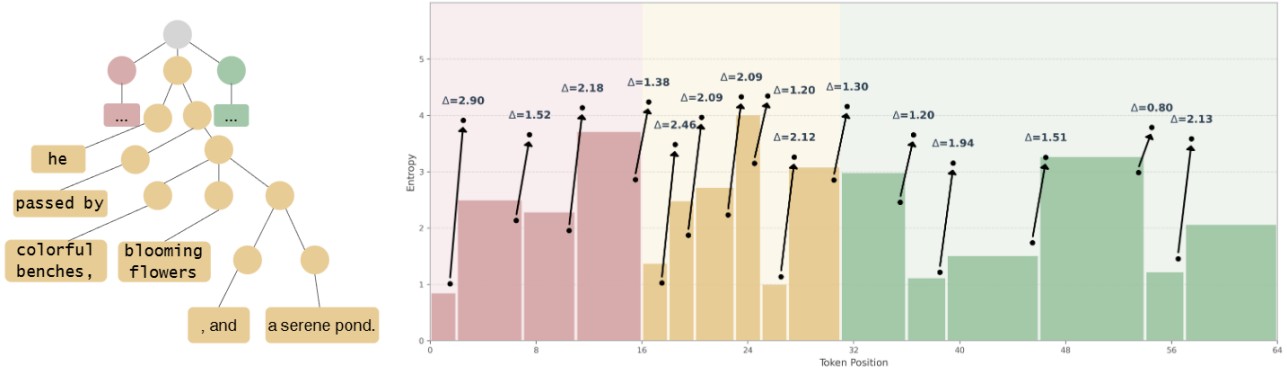

*Figure 2.* Visualization of the alignment between entropy-driven block partitioning and the constituency-parse structure. The left panel shows the constituency parse tree obtained by a conventional syntactic parser, which serves as an external linguistic reference for validation. The right panel shows that entropy-shift-based boundaries are well aligned with semantic and syntactic constituents, supporting the effectiveness of constituent-aware block partitioning.

Combined with KV Cache, Swordsman improves accuracy from 77.40% to 81.50% and achieves 8.79× faster inference on GSM8K compared to vanilla LLaDA. Compared with LLaDA-architecture Fast-dLLM, Swordsman achieves an accuracy improvement from 35.59% to 43.90% on Humaneval and delivers up to 8.31% accuracy improvement while maintaining comparable inference speed.

## 2. Related Work

### 2.1. Diffusion Language Models

Diffusion models (Ho et al., 2020; Esser et al., 2024; Shen et al., 2025; Xu et al., 2026; Zhang et al., 2026), as a generative paradigm based on iterative denoising, have been successfully applied in natural language processing as diffusion language models (DLMs). Among DLMs, masked diffusion models (MDMs)(Sahoo et al., 2024) emerge as the mainstream paradigm by surpassing alternatives(Austin et al., 2021; Lou et al.) through simplified and efficient training objectives. Building upon MDMs, REMDM (Wang et al., 2025a) enhances performance via a remasking sampler for iterative inference refinement, while A-CFG (Li et al., 2025a) balances generation diversity and prompt fidelity through dynamic low-confidence masking. However, these methods fail to address critical bottlenecks in inference latency and generation quality. Inspired by scaling laws, frameworks like LLaDA (Nie et al., 2025) and Dream (Ye et al., 2025) have further unlocked DLM potential through architectural adjustments and parameter scaling, becoming leading foundation models. Building on these advances, recent studies have also extended DLMs to the multimodality, unifying image and text modalities within a diffusion-based framework (You et al., 2025; Yang et al., 2026; You et al., 2026; Li et al., 2026a; Xin et al., 2025; Li et al., 2026b; Yu et al., 2025; Tian et al., 2025).

### 2.2. Efficient inference for DLMs

Recently, researchers have improved LLaDA from various perspectives. LongLLaDA (Liu et al., 2026) enables long-context generation capabilities through a training-free NTK-based RoPE scaling method. *d1* (Zhao et al., 2026) enhances reasoning capabilities via a two-stage post-training framework that combines supervised fine-tuning and reinforcement learning. LLaDA 1.5 (Zhu et al., 2025a) incorporates VRPO-based reinforcement learning for better human preference alignment. Despite progress in specific areas, these methods fail to address LLaDA's core challenges: high inference latency and suboptimal generation quality, limiting its broader applications.

Several methods have been proposed to address these challenges. PC-Sampler (Huang et al., 2026) improves quality by adjusting token decoding order. APD (Israel et al., 2025) accelerates inference via a small autoregressive model. Other studies (Wu et al., 2026b; Nguyen-Tri et al., 2025; Wei et al., 2025) primarily focus on maximizing inference acceleration. However, these methods often involve tradeoffs between speed and quality, leaving a need for a simple and general decoding framework that improves both.

Thus, block-wise decoding (Wu et al., 2025b; Arriola et al., 2025) has emerged to address these efficiency concerns by organizing text into blocks. Decoded blocks can provide cached information for subsequent blocks, thereby accelerating inference (Kim et al., 2025; Wang et al., 2025b). Mean-

while, block-by-block decoding prevents premature generation of unnatural content (i.e., [EOF])(Zhu et al., 2025a). With advantages in both latency and quality, block-wise methods have become the mainstream solution (Zhu et al., 2026; Wu et al., 2025a; Bao et al., 2025). However, existing methods typically use fixed block sizes (Hong et al., 2025; Agrawal et al., 2025), ignoring semantic constituents within sentences. This causes abrupt constituent boundaries and internal confusion within blocks. Current preliminary explorations attempt to partition blocks using punctuation marks(Lu et al., 2025), but such coarse-grained segmentation offers limited efficiency improvement. To address this, we develop a training-free fine-grained partitioning framework that organizes blocks by semantic constituents, maintaining low latency while improving generation quality.

# 3. Methodology

## 3.1. Preliminary

**Decoding Process of DLMs.** Diffusion Language Models (DLMs) generate text through a masked diffusion paradigm (Nie et al., 2025). Starting from a fully masked sequence $\mathbf{x}^{(T)}$ where all tokens are [MASK], the model progressively unmasks tokens over $T$ steps. At each step $t \in \{T, T-1, \ldots, 1\}$ with the updated sequence $\mathbf{x}^{(t)}$, the decoding process is modeled as:

$$p_\theta(\mathbf{x}^{(t-1)}|\mathbf{x}^{(t)}) = \prod_{i \in \mathcal{M}(t)} p_\theta(x_i|\mathbf{x}^{(t)}), \quad (1)$$

where $\mathcal{M}(t)$ contains masked positions at step $t$ and $p_\theta$ denotes the learned decoding transition. Common confidence-based samplers unmask tokens using threshold $\tau$:

$$c_i^{(t)} \triangleq \max_{v \in \mathcal{V}} p_\theta(v \mid \mathbf{x}^{(t)}, i),$$
$$\mathcal{U}^{(t)} = \left\{ i \in \mathcal{M}(t) \mid c_i^{(t)} \geq \tau \right\}, \quad (2)$$

where $\mathcal{V}$ denotes the vocabulary, $c_i^{(t)}$ is the confidence score of token $i$, and $\mathcal{U}^{(t)}$ represents the set of tokens selected for unmasking at step $t$. The sequence $\mathbf{x}^{(t)}$ is then updated as:

$$x_i^{(t-1)} = \begin{cases} \operatorname*{argmax}_{v \in \mathcal{V}} p_\theta(v \mid \mathbf{x}^{(t)}, i), & \text{if } i \in \mathcal{U}^{(t)}, \\ x_i^{(t)}, & \text{otherwise,} \end{cases} \quad (3)$$

with the masked set updated as $\mathcal{M}^{(t-1)} = \mathcal{M}^{(t)} \setminus \mathcal{U}^{(t)}$. This iterative process continues until the masked set is empty, i.e., $\mathcal{M}^{(t)} = \emptyset$, resulting in the final generated sequence $\mathbf{x}^{(0)}$.

**Block-wise DLMs.** Block-wise decoding (Wu et al., 2025b; Arriola et al., 2025) partitions the sequence into $K$ disjoint blocks, denoted as $\mathcal{B} = \{B_1, B_2, \ldots, B_K\}$ and decodes them sequentially. At each step $t$, only tokens within the current block $B^{(t)}$ are considered for unmasking:

$$S^{(t)} = \{i \mid i \in B^{(t)}\}, \quad (4)$$

where $S^{(t)}$ constricts decoding process to the right positions, preventing premature unnatural content generation for better generation quality while enabling two key mechanisms: KV cache across sequential blocks and parallel decoding within each block, both accelerating inference.

## 3.2. Entropy Analysis

Entropy measures the uncertainty of a probability distribution in information theory (MacKay, 2003). For DLMs, given context $\mathcal{C}_i$, the entropy of the $i^{th}$ token $t_i$ is defined as the Shannon entropy of the predictive distribution:

$$H_i = -\sum_{v \in \mathcal{V}} p_\theta(t_i = v \mid \mathcal{C}_i) \log p_\theta(t_i = v \mid \mathcal{C}_i), \quad (5)$$

where $v \in \mathcal{V}$ denotes a candidate token. We utilize the entropy shift $\Delta H_i = H_{i+1} - H_i$ to capture the dynamic changes of uncertainty. Formally, a text sequence $\mathbf{x}$ is composed of multiple independent semantic constituents $\{\mathcal{G}_1, \mathcal{G}_2, \ldots, \mathcal{G}_K\}$. Tokens within constituents are tightly coupled syntactically and semantically, forming strong correlations, while different constituents exhibit weak dependencies due to their distinct semantic expressions.

In DLMs, entropy quantifies the scale of the effective token search space rather than the full vocabulary. We acknowledge that token distributions are typically non-uniform in practice. To make this tractable, we consider the effective candidate vocabulary $V_{\text{eff}}(i)$ of token $t_i$, which contains the tokens with non-negligible probabilities. Within this candidate vocabulary, the distribution can be approximated as locally flat for analytical purposes. Denoting its size as $N_i = |V_{\text{eff}}(i)|$, the entropy of $t_i$ can be modeled as:

$$H_i \approx \log N_i + \epsilon_{\text{flat},i}, \quad (6)$$

where $\epsilon_{\text{flat},i}$ is a shape correction coefficient compensating for the deviation between the actual non-uniform distribution and the locally uniform approximation over $V_{\text{eff}}(i)$. According to Equation 5, the entropy shift $\Delta H_i$ is determined by the change in effective candidate vocabulary sizes of adjacent tokens, together with the residual correction term:

$$\Delta H_i = H_{i+1} - H_i \approx \log \frac{N_{i+1}}{N_i} + (\epsilon_{\text{flat},i+1} - \epsilon_{\text{flat},i}), \quad (7)$$

indicating that vocabulary size differences provide the dominant signal of entropy shifts, while $\epsilon_{\text{flat}}$ accounts for the residual effect of non-uniform distribution shapes.

**Intra-constituent Smoothness.** Within a semantic constituent, adjacent positions share consistent syntactic and semantic constraints from the decoded prefix. Since they belong to the same incomplete semantic constituent, the candidate vocabulary size exhibits smooth variation. We formalize this as the **Intra-Constituent Smoothness Assumption**: for adjacent token $t_i, t_{i+1} \in \mathcal{G}_\mathcal{A}$, the relative

change in candidate vocabulary size satisfies:

$$\left| \frac{N_{i+1} - N_i}{N_i} \right| = |\xi| \leq \delta, \;\; \delta \in [0, 1), \tag{8}$$

where $\delta$ is a constant representing the local prediction uncertainty. When $\delta$ is small, applying the Taylor expansion for $\log(1 + x) \approx x$ yields the upper bound of entropy shift within the same constituent as:

$$|\Delta H_i^{\text{intra}}| \approx |\log(1 + \xi)| \approx |\xi| \leq \delta. \tag{9}$$

This proves that within semantic constituents, entropy shifts are constrained to a small neighborhood.

**Constituent Boundary Abruptness.** When decoding transitions from position $i \in \mathcal{G}_A$ to $i + 1 \in \mathcal{G}_B$, the local strong constraints imposed by $\mathcal{G}_A$ terminate, while those of $\mathcal{G}_B$ remains unestablished. The prediction space degrads from specific constituent expression $N_{\text{local}}$ to general semantic text $N_{\text{global}}$, where the ratio $\rho$ meets:

$$\rho = \frac{N_{\text{global}}}{N_{\text{local}}} \gg 1. \tag{10}$$

With Equation 6, the boundary entropy shift can be put as:

$$\Delta H_i^{\text{boundary}} = H_{i+1} - H_i \approx \log \frac{N_{\text{global}}}{N_{\text{local}}} = \log \rho. \tag{11}$$

Since $N_{\text{global}}$ spans a wide range of syntactic and semantic contexts, it's clear that $\rho$ is substantially large.

Combining the above analysis, the feasibility of semantic boundary detection depends on the following ratio:

$$\frac{\Delta H_i^{\text{boundary}}}{|\Delta H_i^{\text{intra}}|} \approx \frac{\log \rho}{\delta}, \tag{12}$$

when semantic constituents maintain internal coherence ($\delta$ small) and exhibit distinct semantic domains($\rho$ large), we have $\Delta H_i^{\text{boundary}} \gg |\Delta H_i^{\text{intra}}|$. Therefore, detecting local maxima of entropy shifts enables accurate identification of semantic constituent boundaries.

### 3.3. Swordsman

Building upon the entropy analysis in Section 3.2, we present Swordsman, a training-free block-wise decoding framework that leverages entropy shifts to achieve constituent-aware generation in DLMs. As illustrated in Figure 3, Swordsman employs entropy-driven adaptive partitioning to align block boundaries with semantic constituents during generation. Subsequently, each partitioned block is decoded via difficulty-aware parallel unmasking with dynamic thresholds tailored to block-specific uncertainty.

**Adaptive block partition** In the text sequence generation task of DLMs, block-wise decoding models inherently require partitioning the sequence into blocks to enable progressive, block-by-block generation. This partitioning is not merely a technical detail but fundamentally determines the quality of decoding: improper boundaries split semantic constituents, significantly degrading generation performance. In contrast to traditional methods that employ rigid partitioning with predetermined fixed-length blocks before inference, we propose adaptive partitioning that divides block boundaries based on the local maximum entropy shifts $\Delta H_i$ between adjacent tokens during decoding, which reveal the natural transitions between semantic constituents.

Specifically, we employ an iteratively partitioning strategy that progressively determines block boundaries once prefix blocks are decoded. At each iteration after the $k^{th}$ block $B_k$ is decoded, the previously decoded content $\{B_1, \ldots, B_k\}$ provides essential contextual information that refines the model's uncertainty estimates for the remaining masked positions. We re-perform forward propagation on the residual sequence to compute the updated predictive entropy $H_i$ at each remaining position, calculate the entropy shifts $\Delta H_i$ based on these updated values, and identify the next maximum shift point as the right boundary for current block:

$$position_r = \begin{cases} \underset{t < i \leq L}{\arg\max} \, \Delta H_i & \text{if } \underset{t < i \leq L}{\max} \, \Delta H_i \geq \tau_{\min}, \\ L & \text{otherwise}, \end{cases} \tag{13}$$

where $L$ denotes the predefined generation length. As decoding progresses and contextual information accumulates, the uncertainty of remaining tokens diminishes, reflected in smaller entropy shifts. To prevent inefficient oversegmentation in the tail region where uncertainty has substantially converged, we introduce an adaptive termination mechanism with a minimum shift threshold $\tau_{min} = 0.1$, below the entropy shift at semantic constituent boundaries. When the maximum entropy shift in the remaining sequence falls below this threshold, we terminate further partitioning and merge all left tokens into the current block, thereby avoiding unnecessary computational overhead from excessive partitioning while maintaining decoding efficiency.

This entropy-driven partitioning yields two key advantages: intra-block semantic consistency reduces prediction uncertainty within constituents, while inter-block semantic independence allows efficient KV cache reuse without costly out-of-block re-computations. These properties jointly contribute to improved generation quality and reduced inference latency relative to fixed partitioning strategies.

**Dynamic parallel unmask** Due to the precise semantic-based partitioning, tokens within each block are tightly coupled in syntax and semantics, while blocks exhibit significant variations in both scope and difficulty. However,

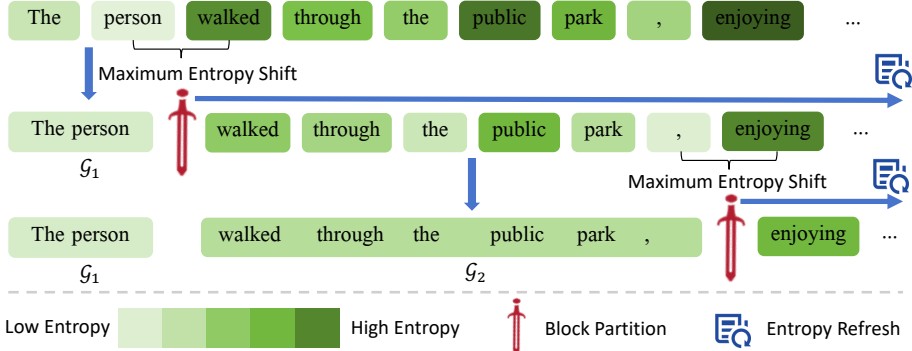

*Figure 3.* Entropy-driven adaptive partitioning for semantic constituents: each step splits at the maximum entropy shift, decodes the block with a dynamic threshold, refreshes entropy, and repeats.

existing static decoding schemes apply a constant confidence threshold across all blocks. This causes severe mismatches: strict thresholds force unnecessary serial decoding in confident blocks, while lenient thresholds enable risky parallelization in uncertain blocks. Such difficulty-agnostic threshold settings ignore the collaboration of tokens within semantic constituents and fail to flexibly adapt decoding strategies to varying block difficulty.

To further leverage the acceleration benefits brought by semantic priors, we propose a dynamic threshold unmask mechanism based on block average entropy as put:

$$\bar{H}_k = \frac{1}{|B_k|} \sum_{t_i \in B_k} H_i. \quad (14)$$

Before decoding each block, we obtain the overall uncertainty of the $k^{th}$ block $B_k$ by calculating its initial average entropy $\bar{H}_k$. High entropy indicates semantic complexity requiring cautious decoding, while low entropy enables aggressive parallelization in the block decoding. This motivates our dynamic threshold unmask mechanism: adapting the confidence threshold to block-specific difficulty enables both safe decoding for uncertain blocks and aggressive parallelization for confident blocks.

Furthermore, we calibrate the $k^{th}$ block's difficulty using historical entropy from preceding blocks as:

$$\lambda_k = 1 - \frac{\bar{H}_k}{\mathcal{H}_{max}^{(k)}}, \quad \mathcal{H}_{max}^{(k)} = \max_{i=1}^{k} \bar{H}_i, \quad (15)$$

Based on the calibrated difficulty metric $\lambda_k$, we compute the dynamic confidence threshold for the block $B_k$ as:

$$\tau_t = \tau_{init} \cdot \left[ (1 - \lambda_k) + \lambda_k \cdot \sqrt{\frac{\bar{H}_t^{(k)}}{\bar{H}_{start}^{(k)}}} \right], \quad (16)$$

where $\tau_{init}$ denotes the global base threshold, while $\bar{H}_t^{(k)}$ and $\bar{H}_{start}^{(k)}$ represents the average entropy of block $B_k$ at the current decoding step $t$ and the initial step respectively.

This dynamic mechanism enables two-level adaptive thresholding. At the inter-block level, the coefficient $\lambda_k$ modulates the threshold based on block-specific difficulty. High-entropy blocks use stricter thresholds to ensure accuracy, while low-entropy blocks use relaxed ones to maximize parallelization. At the intra-block level, the ratio $\sqrt{\frac{\bar{H}_t^{(k)}}{\bar{H}_{start}^{(k)}}}$ captures the entropy decay as decoding progresses within the block. As more tokens are decoded, $\bar{H}_t^{(k)}$ decreases, causing $\tau_t$ to gradually decline. This progressive relaxation allows the model to perform strict selection initially, but gradually ease constraints for better parallelization.

These two levels of threshold adaptation work complementarily: inter-block thresholding matches decoding strategies to semantic constituents' difficulties, while intra-block adjustment dynamically exploits decreasing uncertainty within each constituent. This maximizes inference acceleration while ensuring generation quality, achieving a dynamic balance between speed and quality.

## 4. Experiments

### 4.1. Experimental Setup

**Implementation Details** Our method is evaluated on three pretrained DLMs: LLaDA-8B-Instruct (Nie et al., 2025), LLaDA-1.5 (Zhu et al., 2025a), and Dream-v0-Base-7B (Ye et al., 2025). All experiments are conducted on an NVIDIA H200 GPU. We compare our proposed method against three baselines, including two fixed block methods: Fast-dLLM (Wu et al., 2025b) and D2F (Wang et al., 2025b); and one adaptive block partitioning strategy AdaBlock (Lu et al., 2025). Furthermore, following Fast-dLLM (Wu et al., 2025b), we evaluate the methods across three KV cache configurations: *No cache*, *Prefix cache*, and *Dual cache*. *No cache* disables KV reuse and recomputes all attention states at each decoding step. *Prefix cache* reuses the KV states of previously decoded prefix blocks to reduce redundant

computation. *Dual cache* further caches both prefix-side and suffix-side states in block-wise bidirectional decoding, enabling more extensive KV reuse for DLM inference. For all tasks, we set the generation budget length to $L = 512$, and the static confidence threshold $\tau_{static} = 0.9$. Swordsman introduces two hyperparameters: the minimum entropy shift $\tau_{min} = 0.1$ of the adaptive block partition strategy and the base threshold $\tau_{init} = 0.9$ of dynamic parallel unmask mechanism. For fixed block methods, the block size is set to 32 (the optimal set identified by Fast-dLLM).

*Table 1.* Generation quality comparison of different models with various cache configurations on four benchmarks.

| Experiment Setting | | | Accuracy ⇑ | | | |
|---|---|---|---|---|---|---|
| Partition | Method | Cache | GSM8K | MATH | Humaneval | MBPP |
| **LLaDA-8B-Instruct** | | | | | | |
| Fixed | Fast-dLLM | None | 77.56 | 36.52 | 43.90 | 14.20 |
| | | Prefix | 77.10 | 36.22 | 41.46 | 13.20 |
| | | Dual | 75.21 | 35.46 | 44.51 | 13.60 |
| | D2F | Prefix | 74.98 | 28.76 | 36.59 | 38.00 |
| Adaptive | AdaBlock[†] | None | 80.60 | **37.30** | 43.30 | **39.80** |
| | | Dual | 78.50 | 35.30 | **46.30** | 38.00 |
| | AdaBlock | None | 80.06 | **37.30** | 43.30 | 14.20 |
| | | Dual | 76.80 | 35.16 | 45.27 | 11.40 |
| | Swordsman | None | 81.43 | 36.82 | 42.68 | 13.00 |
| | | Prefix | 80.67 | 36.68 | 43.90 | 13.60 |
| | | Dual | **81.50** | 35.76 | 44.51 | 13.60 |
| **Dream-v0-base-7B** | | | | | | |
| Fixed | Fast-dLLM | None | 75.97 | **40.08** | 52.44 | 55.40 |
| | | Prefix | 74.00 | 38.96 | 56.70 | **55.80** |
| | | Dual | 74.75 | 38.34 | 53.05 | 54.40 |
| | D2F | Prefix | 76.12 | 38.62 | 52.55 | 53.48 |
| Adaptive | AdaBlock[†] | None | 75.70 | 39.90 | 51.20 | 14.20 |
| | | Dual | 75.10 | 38.40 | 53.00 | 11.60 |
| | AdaBlock | None | 75.59 | 39.46 | 51.20 | 0.00[‡] |
| | | Dual | 75.12 | 38.47 | 52.61 | 0.00[‡] |
| | Swordsman | None | 75.82 | 40.00 | 54.27 | 55.60 |
| | | Prefix | **76.88** | 39.42 | **57.93** | **55.80** |
| | | Dual | 76.50 | 38.58 | 55.49 | 54.80 |
| **LLaDA-1.5** | | | | | | |
| Fixed | Fast-dLLM | None | 82.56 | **37.18** | 39.02 | 39.60 |
| | | Prefix | 81.80 | 34.60 | 39.02 | 40.00 |
| | | Dual | 80.52 | 33.26 | 35.59 | 36.20 |
| Adaptive | AdaBlock[†] | None | 82.40 | 36.70 | 38.40 | 37.60 |
| | | Dual | 81.70 | 33.90 | 39.00 | 36.40 |
| | AdaBlock | None | 82.03 | 36.56 | 38.42 | 37.80 |
| | | Dual | 82.18 | 33.74 | 39.17 | 37.00 |
| | Swordsman | None | **84.00** | 36.58 | 42.68 | **41.00** |
| | | Prefix | 82.56 | 36.94 | 42.02 | 39.80 |
| | | Dual | 82.87 | 35.30 | **43.90** | 39.40 |

*Note.* [†] denotes results quoted from the original paper of AdaBlock, as the code was not publicly available during our initial comparison and our reproduction showed large discrepancies. After its release, we further evaluated the official implementation under the unified Fast-dLLM protocol used in our evaluation without post-processing. [‡] indicates that the released AdaBlock implementation achieves zero accuracy on MBPP under this protocol.

**Benchmarks and Metrics.** We evaluate on standard LLM

benchmarks covering two categories: (1) code generation, including HumanEval (0-shot) and MBPP (3-shot); (2) mathematical reasoning, including GSM8K (5-shot) and MATH (4-shot). For evaluation metrics, we report **Accuracy** (%, ⇑) for generation quality, **Throughput** (TPS, ⇑) for decoding speed, and **Latency** (s/sample, ⇓) for end-to-end generation time. Throughout the tables, **bold** and underline denote the best and second-best results, respectively.

### 4.2. Main Results

**Overall Performance**   Table 1 presents the comprehensive comparison across three DLMs and four benchmarks. Swordsman achieves **state-of-the-art** performance across multiple models and benchmarks, validating the efficiency of our entropy-driven partitioning approach. Specifically, on LLaDA-8B-Instruct, Swordsman achieves 81.50% on GSM8K, surpassing Fast-dLLM by +6.29% and the officially released AdaBlock by +1.44% under the unified evaluation protocol. On LLaDA-1.5, Swordsman attains the best performance on GSM8K, HumanEval, and MBPP, notably reaching 84.00% on GSM8K (+1.44% over Fast-dLLM and +1.82% over AdaBlock) and 43.90% on HumanEval (+8.31% over Fast-dLLM with Dual cache). For Dream-v0-base-7B, Swordsman yields the highest GSM8K accuracy of 76.88% and the highest HumanEval accuracy of 57.93%, while also achieving the tied-best MBPP result of 55.80%. These improvements validate our core hypothesis: entropy-driven adaptive partitioning better captures the natural structure of semantic constituents, enabling more efficient parallel decoding. For completeness, we include both the originally reported AdaBlock results and the results obtained from its released implementation under our unified evaluation protocol. The MBPP discrepancy further motivates our protocol-consistent comparison, with additional analysis provided in Appendix A.

**Analysis of Inference Speed Performance**   We further analyze the inference speed across DLMs in Table 2. Under the dual cache configuration, Swordsman consistently improves TPS over Fast-dLLM by +1.62, +1.48, and +3.67 on LLaDA-8B-Instruct, Dream-v0-base-7B, and LLaDA-1.5, respectively, while reducing latency by 0.27s on average. Compared with the released AdaBlock implementation, Swordsman also achieves higher Dual-cache throughput across all three backbones, with TPS gains of +1.34, +2.56, and +3.26, and consistently lower latency. Although D2F attains the highest TPS on Dream-v0-base-7B and the lowest latency on two backbones, it suffers clear accuracy degradation shown in Table 1. Overall, Swordsman provides a stronger training-free speed-quality trade-off by improving inference efficiency through adaptive block partitioning, dynamic parallel unmasking without model retraining. These results indicate that adaptive block partitioning does not

*Table 2.* Inference speed comparison of different models with various cache configurations. All results are measured with batchsize = 1 on the 1,319 GSM8K samples.

| Method | Cache | LLaDA-8B-Instruct | | Dream-v0-base-7B | | LLaDA-1.5 | |
| | | TPS ⇑ | Latency ⇓ | TPS ⇑ | Latency ⇓ | TPS ⇑ | Latency ⇓ |
|---|---|---|---|---|---|---|---|
| Fast-dLLM | None | 42.36 | 6.61 | 39.98 | 12.54 | 40.43 | 5.23 |
| | Prefix | 58.42 | 4.59 | 62.75 | 8.21 | 50.16 | 3.90 |
| | Dual | 72.12 | 3.72 | 74.37 | 7.11 | 61.30 | 3.41 |
| D2F | Prefix | 64.49 | **2.24** | 89.82 | **4.32** | – | – |
| AdaBlock* | None | 45.02 | 5.98 | 43.12 | 11.64 | 41.06 | 4.92 |
| | Dual | 58.21 | 4.73 | 63.73 | 8.14 | 54.69 | 3.16 |
| AdaBlock | None | 44.42 | 6.06 | 43.08 | 11.89 | 41.38 | 4.91 |
| | Dual | 72.40 | 3.92 | 73.29 | 7.03 | 61.71 | 3.23 |
| Swordsman | None | 44.89 | 6.00 | 42.34 | 12.07 | 41.14 | 4.89 |
| | Prefix | 58.68 | 4.54 | 63.88 | 7.99 | 52.55 | 3.69 |
| | Dual | **73.74** | 3.66 | 75.85 | 6.74 | **64.97** | **3.03** |

*Note.* * denotes our pre-release reproduction based on the paper description of AdaBlock. After it was released, we further evaluated its official implementation under the same inference setting.

introduce extra inference overhead; instead, it improves the decoding schedule by enabling more effective parallel unmasking and cache reuse.

**Robustness across Cache Configurations** Despite the fact that caching strategies introduce computational approximations that can degrade accuracy, as evidenced by Fast-dLLM's performance drop from 77.56% (None) to 75.21% (Dual) on GSM8K for LLaDA-8B-Instruct, Swordsman maintains consistently superior performance across all cache configurations, delivering stable gains over Fast-dLLM. Aggregating results across the four benchmarks, Swordsman surpasses Fast-dLLM under every cache setting with an average improvement of about +1.50% and the advantage becomes more pronounced when stronger caching is used, specifically on HumanEval under Dual caching for LLaDA-1.5, where Swordsman achieves 43.90% but 35.59% for Fast-dLLM, an absolute gain of +8.31%, highlighting Swordsman's markedly higher robustness to cache-induced approximation in code-generation settings. Across the entire datasets, Swordsman consistently achieves substantial speedup from the caching strategy according to Table 2 compared to the non-caching performance.

**Analysis of Efficiency** Figure 4 compares efficiency across cache configurations and backbones. Swordsman consistently delivers the best or near-best accuracy across all settings, with stable gains over Fast-dLLM. On GSM8K, it reaches 81.50% on LLaDA-8B-Instruct under Dual cache (vs. 75.21% for Fast-dLLM) and attains the overall peak of 84.00% on LLaDA-1.5 with None cache, while maintaining 82.87% under Dual. Crucially, these accuracy gains do not sacrifice speed. Averaged over all comparable settings on GSM8K, Swordsman yields +1.79 higher throughput and -0.3s lower latency than Fast-dLLM, with a +2.53% accuracy improvement. Under Dual caching specifically, it achieves a larger gain of +2.20 TPS, -0.27s latency, and

+3.46% accuracy with Fast-dLLM. This joint superiority is clearly reflected in the Pareto analysis: four Swordsman configurations lie on the frontier, each achieving superior speed-accuracy combinations. While D2F also reaches the frontier through training-based distillation, it occupies the extremely high-speed, low-accuracy region. The strongest demonstration of Swordsman's superiority is on LLaDA-1.5 with Dual cache, simultaneously achieving high throughput (64.97 TPS), the lowest latency (3.03s), and strong accuracy (82.87%), demonstrating that entropy-driven block partitioning and dynamic threshold adjustment for parallel unmasking enable both high inference speed and superior generation quality without model retraining.

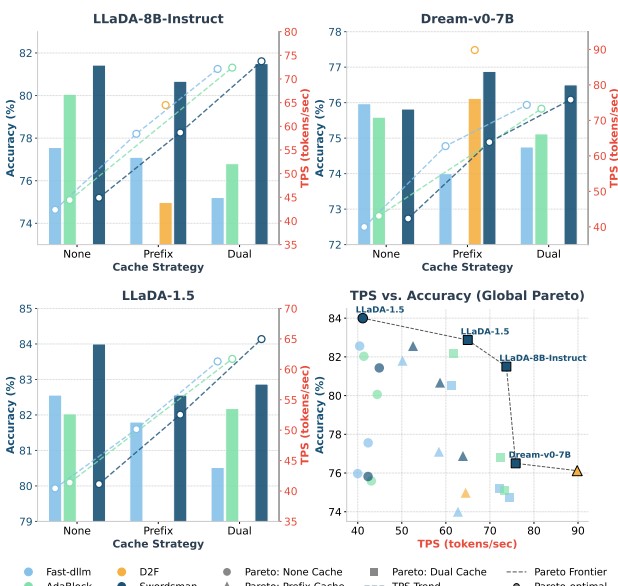

*Figure 4.* Visualization of throughput and accuracy (with the underlying results reported in Tables 1 and 2). The upper-left, lower-left, and upper-right panels compare TPS and accuracy on GSM8K for three base models, using bars for accuracy and markers for TPS, with colors denoting different methods. The lower-right panel presents the global Pareto analysis over all model, method, and cache configurations, where marker shapes indicate cache strategies, and the dashed line marks the Pareto frontier.

**Progressive Gains from Adaptive Block-wise Decoding** Table 3 analyzes how each decoding component contributes

*Table 3.* Progressive contribution of decoding components on GSM8K with LLaDA-8B-Instruct. Accuracy and throughput are reported as block-wise decoding, parallel decoding, and adaptive partitioning are cumulatively added.

| Method | Block-wise | Parallel | Adaptive | Accuracy ⇑ | TPS ⇑ |
|---|---|---|---|---|---|
| LLaDA | ✗ | ✗ | ✗ | 29.04 | 3.32 |
| | ✓ | ✗ | ✗ | 77.40 | 8.39 |
| | ✓ | ✓ | ✗ | 77.56 | 42.36 |
| Swordsman | ✓ | ✓ | ✓ | **81.43** | **44.89** |

to performance on GSM8K with LLaDA-8B-Instruct. Starting from full-diffusion LLaDA (29.04%, 3.32 TPS), introducing block-wise decoding improves accuracy to 77.40% (+48.36%) and increases throughput to 8.39 TPS. Adding confidence-based parallel decoding further raises throughput from 8.39 to 42.36 TPS with almost unchanged accuracy, showing that the block-wise constraint preserves a coarse decoding order, under which intra-block parallelism can accelerate generation without sacrificing quality. Finally, Swordsman adds entropy-driven adaptive partitioning and achieves the best result, improving accuracy to 81.43% while maintaining high throughput (44.89 TPS). This final gain suggests that, by placing boundaries at entropy shifts, adaptive blocks better preserve semantic constituents within blocks and reduce uncertainty during parallel decoding. Overall, the cumulative improvements show that block-wise decoding provides the ordering foundation, parallel decoding brings the main speedup, and adaptive partitioning further improves both quality and efficiency by aligning decoding blocks with the latent semantic structure of the sentence to be unmasked.

### 4.3. Ablation Studies

We conduct extensive experiments to understand how the overall strategies of Swordsman contribute to performance. Unless otherwise stated, all ablation studies are conducted on GSM8K using LLaDA-8B-Instruct, with caching disabled and a maximum generation length of 512 tokens.

**Effect of Dynamic Thresholding**   Figure 5 ablates dynamic thresholding by comparing it with a static threshold and varying its initial value $\tau_{init}$. The static threshold $\tau_{static} = 0.9$ (gray bar) applies a fixed confidence constraint throughout the decoding process. When remaining tokens fail to meet $\tau_{static}$, the model falls into a degradation state (Wu et al., 2025b), where tokens must be decoded sequentially by selecting the highest-confidence token one at a time, significantly slowing down inference speed. Our dynamic threshold adaptively relaxes confidence as tokens

are decoded and information accumulated, enabling more parallel decoding. With the default setting $\tau_{init} = 0.9$, the dynamic threshold achieves comparable accuracy to the static threshold (81.43% vs. 81.45%) while reducing latency by 2.49s. We further vary $\tau_{init}$ to examine its effect. Lower ones (e.g., 0.5) further reduce latency but cause significant accuracy drops due to premature decoding of uncertain tokens. Conversely, $\tau_{init} = 1.0$ is overly strict, resulting in $\sim 2.4\times$ higher latency than the default setting due to excessive sequential decoding. Therefore, $\tau_{init} = 0.9$ provides the best trade-off between accuracy and latency.

**More Details**   We provide more detailed ablation studies in the appendix, including hyperparameter ablations (Appendix B), comparisons with alternative shift metrics (Appendix D), robustness to generation length and decoding order (Appendix D and F), and further analysis of our method's efficiency (Appendix C and E).

## 5. Conclusion

We introduce Swordsman, a training-free framework that achieves constituent-aware block-wise decoding through entropy-driven adaptive partition. By aligning blocks with semantic boundaries via entropy shift detection and modulating decoding strategies through dynamic thresholding, Swordsman addresses the semantic misalignment inherent in fixed block approaches. Experiments show Swordsman improves vanilla LLaDA accuracy from 77.40% to 81.50% on GSM8K with 8.79× speedup, and achieves +8.31% over Fast-dLLM on HumanEval (35.59% → 43.90%) at matched speed. These results validate that entropy-based semantic awareness enables efficient parallel generation. Current limitations include validation primarily on block-wise DLMs and the potential need for dataset-specific hyperparameter tuning. Future work includes extending to semi-autoregressive models, developing adaptive parameter selection, and improving efficiency through learned mechanisms.

## Acknowledgement

This work is supported by the National Natural Science Foundation of China (No. 62576251, No. 62406225, No. 62376198, and No. 62576247), the National Science and Technology Major Project of China (No. 2025ZD0219200), and the Shanghai Science and Technology Committee under Grant No. 24511103900.

## Impact Statement

This paper presents work whose goal is to advance the field of Machine Learning. There are many potential societal consequences of our work, none of which we feel must be specifically highlighted here.

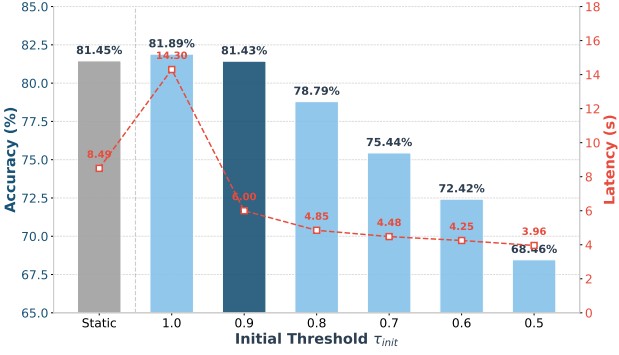

*Figure 5.* Accuracy and latency comparison between static threshold with $\tau_{static} = 0.9$ and dynamic thresholds with varying $\tau_{init}$

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

## A. Additional Analysis on MBPP Anomalies and Protocol Alignment

We provide an additional analysis of the MBPP results in Table 1. During the initial submission, the official implementation of AdaBlock (Lu et al., 2025) was not publicly available. Therefore, we could only quote the results reported in the AdaBlock paper, denoted as AdaBlock[†] in Table 1. These reported results were obtained with AdaBlock's own evaluation pipeline, which applies a post-processing script for code generation evaluation. In contrast, our evaluation follows the Fast-dLLM (Wu et al., 2025b) protocol and directly uses the output from the `lm_eval` framework without additional post-processing. This protocol difference leads to large gaps on MBPP, especially for LLaDA-8B-Instruct, where the reported AdaBlock[†] scores are 39.80% and 38.00%, while its released implementation under our protocol obtains 14.20% and 11.40%.

After AdaBlock was released, we further evaluated its official implementation under the same protocol used for our method and other baselines. This gives the second set of AdaBlock results in Table 1. Under this setting, the MBPP scores of AdaBlock on LLaDA-8B-Instruct become much closer to those of other methods evaluated without post-processing. This suggests that the large MBPP gap between AdaBlock[†] and AdaBlock mainly comes from the evaluation protocol rather than from the block partition strategy alone.

We also observe that AdaBlock obtains 0.00% on MBPP with Dream-v0-Base-7B under our protocol. We suspect that this result may be related to an output-format mismatch when the released AdaBlock implementation is evaluated directly with `lm_eval` and without its post-processing script. For transparency, we keep this result in Table 1 and mark it in the table note. Overall, these observations motivate our protocol-aligned comparison: all reproduced results are evaluated with the same Fast-dLLM protocol and without method-specific post-processing, so the reported differences more directly reflect the decoding and partitioning methods.

## B. Analysis of Minimum Shift Threshold

As shown in Figure 6, when the threshold is set between 0.001 and 0.1, the accuracy remains stable around 81.43%. As it becomes large ($\geq 0.2$), accuracy drops, which indicates that an overly large threshold causes the model to miss critical semantic constituent boundaries. On the other hand, setting the threshold too small forces the model to find splitting points even in regions where shifts are small, unnecessarily separating highly confident tokens into different blocks and increasing decoding steps, which introduces a marginal latency overhead without yielding quality benefits. Empirically, our method demonstrates stable accuracy across a wide range of threshold values (0.001-0.1), indicating robustness to hyperparameter selection. Among these configurations, we identify an optimal setting that achieves the best balance between inference speed and generation quality.

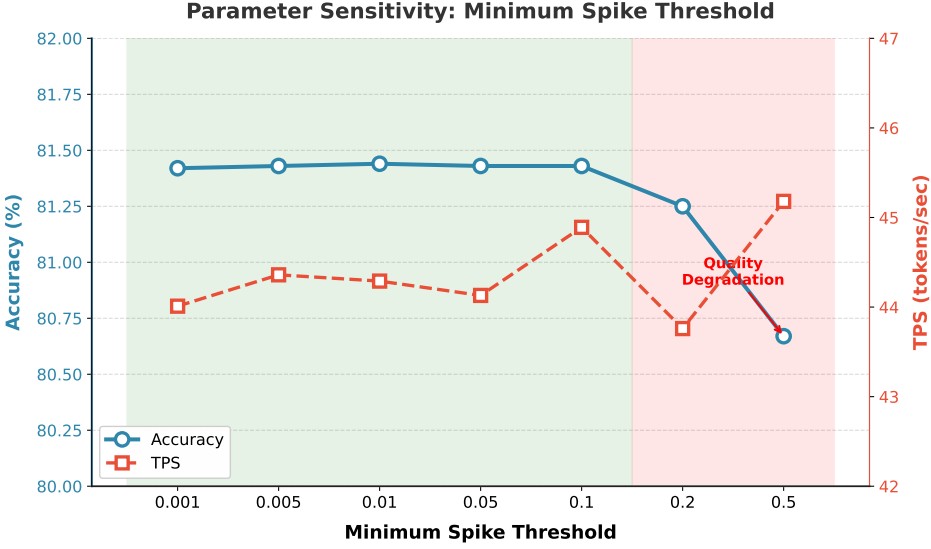

*Figure 6.* Analysis of minimum shift threshold

## C. Effect of Decoding Order

To investigate whether the performance gain of Swordsman stems from constituent-aware partition or the implicit decoding order they impose, we design a composed block ablation. Specifically, we take Swordsman's partitioning results and merge consecutive semantic blocks (e.g., $B_A$, $B_B$, $B_C$, $B_D$) into a single block $B_{composed}$ of approximately 32 tokens as shown in Figure 7. Within this composed block, tokens are decoded by confidence threshold without respecting original boundaries, effectively disrupting the left-to-right order across semantic constituents. This modification leads to an accuracy drop from 81.43% to 78.37%. The degradation can be attributed to the lack of prefix information: in Swordsman, block $B_D$ is decoded only after $B_A$, $B_B$, and $B_C$ are complete, ensuring full preceding context; in $B_{composed}$, tokens from $B_D$ may be decoded while $B_A$, $B_B$, $B_C$ remain partially unresolved, forcing the model to predict with incomplete information. This increases uncertainty for later-position tokens and leads to error accumulation. The results confirm that Swordsman's semantic boundaries serve more than capturing coherent semantic constituents, but implicitly enforcing a general left-to-right generation order, both essential for high-quality generation in diffusion language models.

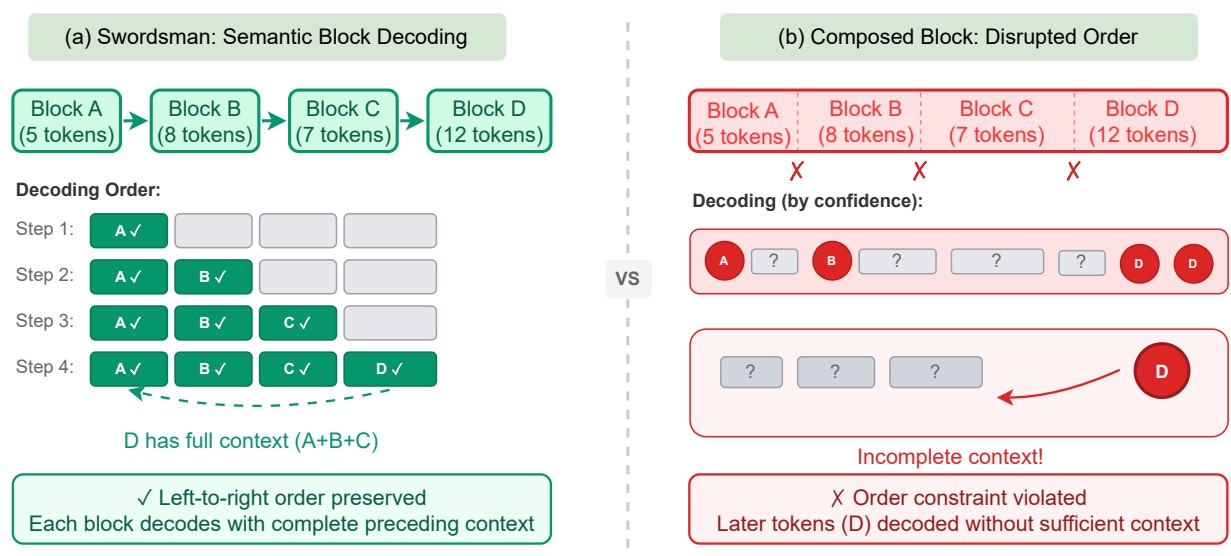

*Figure 7.* Analysis of decoding order

## D. Comparisons Between Metrics and Generation Lengths

We compare several token-level uncertainty metrics for semantic constituent boundary detection and block partition, replacing Shannon entropy with Varentropy, Gini Impurity, Margin, and Confidence. Each metric yields a score for every token, we then compute the score differences between adjacent tokens and divide boundaries at the salient shift points. As shown in Figure 8, entropy achieves the best accuracy, suggesting it provides the most informative signal for locating semantic constituent boundaries among the metrics considered. We further evaluate robustness to the maximum generation length by varying it among {256, 512, 1024}. As shown in Table 4, Swordsman consistently outperforms Fast-DLLM across all lengths and exhibits robust performance over diverse generation lengths.

*Table 4.* Accuracy under different generation lengths for different models.

| Model | GenLen | | |
|---|---|---|---|
| | 256 | 512 | 1024 |
| Swordsman | 79.61 | **81.43** | 79.15 |
| Fast-dLLM | 77.79 | 77.56 | 77.30 |

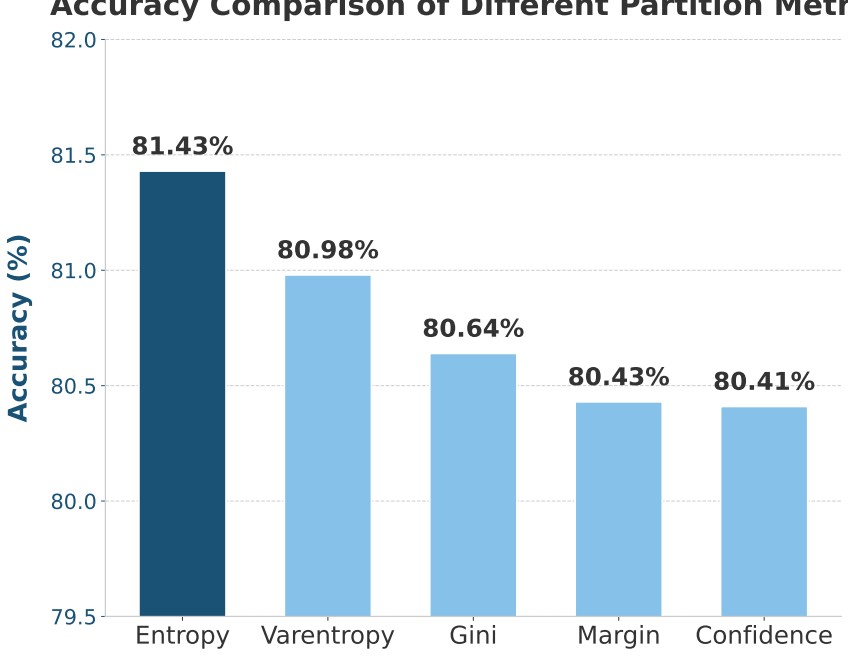

*Figure 8.* Accuracy comparison of different partition metrics.

## E. Effect of Minimum Block Size Constraint

We evaluate whether enforcing a minimum block size mitigates overly fragmented segmentation. As shown in Figure 9, Swordsman performs best (81.43% accuracy). Increasing the minimum block size to $m \in \{4, 8, 16\}$ uniformly reduces accuracy by 1.12% to 2.47%, while TPS improves only marginally. This suggests that length-based constraints can misalign block boundaries with semantic constituents: to satisfy the minimum-length requirement, semantically unrelated adjacent constituents are forced to merge, leading to decoding under incomplete or mixed semantic context. These results support our hypothesis that constituent boundaries should be induced by entropy shifts rather than imposed by arbitrary length constraints.

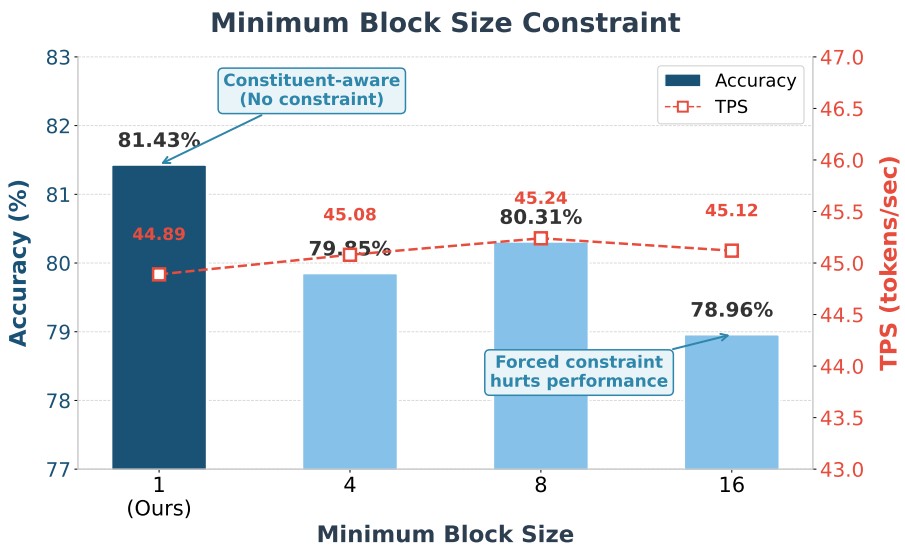

*Figure 9.* Comparison across different minimum block sizes

# F. Robustness to Decoding Strategies

We further verify that our adaptive block partitioning consistently outperforms fixed block partitioning regardless of the decoding strategy, where the parallelization is permitted or not. As shown in Figure 10, under fair comparison with identical threshold parameters $\tau_{static} = \tau_{init} = 0.9$, Swordsman achieves +3.52% and +3.89% accuracy improvements over Fast-dLLM under Top-1 Confidence and Confidence Threshold decoding, respectively. This demonstrates that the advantage of our method stems from better constituent-aware block partition rather than the choice of decoding strategy.

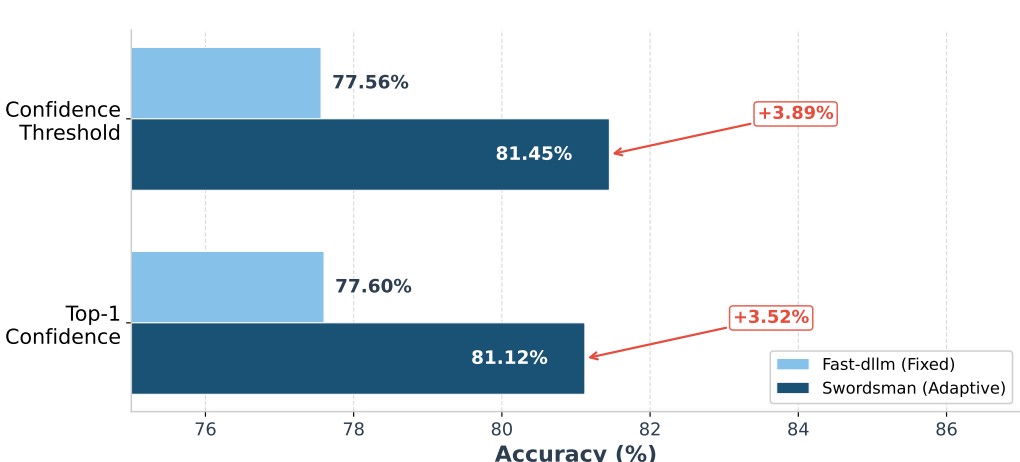

*Figure 10.* Analysis of decoding strategy

