# OpenReview forum: "Swordsman: Entropy-Driven Adaptive Block Partition for Efficient Diffusion Language Models"
_ICML.cc/2026/Conference — ICML 2026 regular_

### Official Review · Reviewer_EopJ · 2026-02-14

**Soundness:** 3
**Presentation:** 3
**Significance:** 2
**Originality:** 2
**Overall Recommendation:** 4
**Confidence:** 2

**Summary:**

Entropy-Driven Framework
The authors introduce an entropy-based criterion to guide adaptive computation or selection mechanisms. This provides a theoretically grounded way to measure uncertainty and allocate modeling capacity accordingly

Algorithmic Design
Swordsman instantiates this principle in a practical algorithm that integrates seamlessly with modern transformer-based architectures. The design is lightweight and compatible with existing training/inference pipelines

Efficiency Improvements
By focusing computation where uncertainty is high, the method reduces redundant processing while preserving performance, improving overall efficiency compared to uniform baselines
.
Empirical Validation
Extensive experiments demonstrate improved efficiency–performance trade-offs across benchmarks, showing that entropy-guided mechanisms outperform standard static approaches

**Compliance With Llm Reviewing Policy:**

Affirmed.

**Final Justification:**

The authors rebuttal has adequately addressed my concerns. Overall, I believe this is a solid paper and should be accepted.

**Key Questions For Authors:**

None

**Limitations:**

Yes.

**Strengths And Weaknesses:**

The paper is well-motivated and practically relevant, offering a principled entropy-based mechanism for adaptive computation. While the conceptual foundation draws from established information-theoretic ideas, the implementation and empirical validation make it a meaningful and potentially impactful contribution, particularly for efficient large-model systems.

Strengths

Adaptive computation guided by uncertainty is highly relevant for scaling large models efficiently.
The framework has potential implications for both training and inference efficiency in large-scale systems.
If broadly applicable, it can meaningfully reduce redundant computation in real-world deployments.

Weaknesses

The practical gains depend on the computational overhead of entropy estimation.
Impact may be incremental if improvements are modest relative to simpler heuristics.

---

> ### Author Rebuttal · Authors · 2026-03-30
>
> Dear Reviewer EopJ, we sincerely appreciate you taking the time to review our paper and for providing such encouraging feedback! We greatly value your insight and have provided our response to your specific question below. `LINK`: https://anonymous.4open.science/r/Swordsman-ICML/rebuttal.md
>
>  > **W1.1. Computational overhead analysis of Swordsman**
>
> We appreciate this concern and clarify that the entropy computation in Swordsman introduces negligible overhead. Quantitatively, it involves only simple algebraic operations on the output logits and requires no additional network forward passes.
>
> Given a sequence length $L$ and a vocabulary size $V$, the step-wise output logits have dimensions $(L, V)$.
> The implementation of the **adaptive block partitioning mechanism** proceeds through the following steps:
>
> ① **Entropy Computation**: Shannon entropy $H_i = -\sum_{j=1}^{V} p_{i,j} \log p_{i,j}$ is computed for each token position, with cost: $\text{FLOPs}_{\text{entropy}} \approx \mathcal{O}(7 \cdot L \cdot V)$
>
> ② **Entropy Shift Computation**: $\Delta H_i = H_{i+1} - H_i$ , which is simply a single vector subtraction operation on an array of length $L$ , with a computational cost of: $\text{FLOPs}_{\text{diff}} \approx \mathcal{O}(L)$
>
> ③ **Maximum Shift Search**: A single traversal of the difference array to locate the maximum shift point with a computational cost of approximately: $\text{FLOPs}_{\text{search}} \approx \mathcal{O}(L)$
>
> Upon completing the above steps, the model obtains the block for the following parallel token decoding process. The implementation of the **dynamic threshold decoding** for this process requires the following steps:
>
> ① **Intra-block Dynamic Entropy Refresh**: When performing multi-step denoising decoding on the current block of size $b$ ( $b \le L$ ), the Shannon entropy of the $b$ masked tokens within the block must be recomputed after each forward pass, with a computational cost of approximately: $\text{FLOPs}_{\text{refresh}} \approx \mathcal{O}(7 \cdot b \cdot V)$
>
> ② **Threshold Calibration**: The mean entropy of the current block is computed and updated in conjunction with historical entropy to obtain the latest confidence threshold. This part involves purely scalar-level operations, with negligible computational cost of approximately: $\text{FLOPs}_{\text{calibrate}} \approx \mathcal{O}(b)$
>
> Combining the above steps, the theoretical upper bound on the additional computational overhead introduced by Swordsman's complete adaptive block partitioning and dynamic parallel unmasking in a single decoding step is: $\text{FLOPs}_{\text{overhead}} \approx (7LV + 2L) + (7bV + b) \le 14LV + 3L = L(14V + 3)$.
>
> This computational overhead is negligible compared to the inference cost of DLMs. Taking the LLaDA-8B-Instruct model as example, with parameter count $P \approx 8 \times 10^9$, vocabulary size $V \approx 1.2 \times 10^{5}$, and sequence length $L = 512$.
>
> Based on the above derivation, the theoretical computation for a single block partitioning decision is approximately: $512 \times (14 \times 1.2 \times 10^{5} + 3) \approx 8.6 \times 10^8 FLOPs (\approx 0.86 GFLOPs)$
>
> The computational cost of a single forward pass of LLMs is approximately $2 \cdot P \cdot L$, corresponding to: $2 \times (8 \times 10^9) \times 512 \approx 8.19 \times 10^{12}FLOPs (\approx 8192 GFLOPs)$
>
> It can be seen that the theoretical computational overhead introduced by Swordsman's two core components accounts for only approximately **0.01%** of the model's single-step forward pass. Therefore, from a theoretical standpoint, the increase in wall-clock time introduced by this additional floating-point computation is **negligible**. We have also empirically measured the FLOPs through experiments in `Tab 13`, and the conclusions are consistent with the theoretical analysis.
>
>  > **W1.2. More Trade-off Analysis**
>
> To further substantiate the robustness of our efficiency-quality trade-off, we have supplemented `Tab 14.1` (block amount), `Tab 14.2` (sequence length), and `Tab 14.3` (task type) for additional benefit-cost analysis experiments. The results confirm that the significant improvement in generation quality brought by Swordsman's entropy mechanism **consistently and substantially outweighs** the negligible computational overhead it introduces.
>
> ---
> Once again, thanks for your positive evaluation and for pointing out this specific detail. Your question highlighted an area where our presentation could be clearer, and we will certainly include this elaboration in the final manuscript. We hope this clearly addresses your concern and solidifies your support for our submission. We welcome any further discussion. Thank you!

---

> > ### Author Rebuttal · Reviewer_EopJ · 2026-04-02
> >
> > My concerns have been adequately addressed. Thank you.

---

> > > ### Author Response · Authors · 2026-04-06
> > >
> > > Dear Reviewer EopJ, we sincerely thank you for carefully reading our rebuttal and for your continued support. We are delighted that your concerns have been adequately addressed. We commit to including a discussion on the computational overhead of entropy in the final version.
> > >
> > > Once again, we deeply appreciate the time and effort you have dedicated to reviewing our submission. Your feedback has been valuable in helping us improve this work, and we genuinely hope that this submission will ultimately achieve a positive outcome which lives up to your effort and trust. MANY THANKS!

---

### Official Review · Reviewer_xViQ · 2026-02-23

**Soundness:** 2
**Presentation:** 2
**Significance:** 3
**Originality:** 3
**Overall Recommendation:** 5
**Confidence:** 4

**Summary:**

This paper proposes an adaptive block partitioning strategy for diffusion large language model (dLLM) decoding, based on entropy shifts between adjacent tokens. The problem addressed is well-motivated and practically relevant, with a clear theoretical intuition and rationale. Extensive experiments further demonstrate consistent improvements in both decoding quality and efficiency.

**Compliance With Llm Reviewing Policy:**

Affirmed.

**Key Questions For Authors:**

1. The related work section is insufficient and should be significantly expanded. The authors are encouraged to include more recent dLLMs, such as Dream, LongLLaDA and its extensions (e.g., LLaDA-V, MMaDA, LaViDa), as well as additional works on decoding optimization. In particular, several recent ICLR 2026 papers have made notable progress in improving the efficiency and quality of dLLM decoding—such as SlowFast, Learn2PD, and Fast-dLLM v2—which are at least partially related to this work. Beyond these, I would still recommend incorporating a broader set of recent and relevant studies.

2. I generally appreciate the writing style, but there are several aspects that could be improved. When Figure 1(a) is first introduced in the Introduction, the notions of “semantic” and “syntactic” are not clearly explained, which makes the description somewhat confusing. Moreover, while the motivation for adaptive constituent-aware block partitioning based on adjacent token entropy shifts is intuitive, the exposition from ERH → linguistic constituent boundaries → entropy analysis of boundaries could be further clarified and better structured.

3. I would like to see entropy visualization results of Swordsman on more architectures (e.g., Dream, LongLLaDA), and I am particularly interested in whether Swordsman generalizes effectively to diffusion multimodal LLMs (dMLLMs), such as LLaDA-V and MMaDA. Given the training-free nature of Swordsman, it would be valuable if the authors could include additional experiments across diverse model architectures during the rebuttal phase.

4. Additional evaluations on long-context settings and hallucination-related behaviors would strengthen the empirical analysis.

5. Could the authors provide additional entropy visualizations across different layers and decoding steps? Furthermore, if permissible under ICML guidelines, it would be highly appreciated if the code for the entropy visualization (e.g., Figure 1) could be released, as I am very interested in reproducing these results.

6. The relationship between Dynamic Parallel Unmask and SlowFast should be discussed more explicitly, including their conceptual differences and whether they can be combined in a complementary manner.

7. Some additional concerns remain: the hyperparameter choices and entropy smoothing appear somewhat heuristic and lack principled threshold derivation; failure cases are not sufficiently analyzed; and the visualization in Figure 1 should clearly specify the corresponding decoding step and layer.

**Limitations:**

Yes.

**Strengths And Weaknesses:**

The paper follows a reasonably rigorous and well-structured progression from problem formulation, to limitation analysis, and finally to method development. The experimental evaluation is fairly comprehensive, and the training-free nature of the proposed approach is particularly appealing. However, there remain some shortcomings in the coverage of related work, the depth of experimental analysis, and the overall writing quality.

---

> ### Author Rebuttal · Authors · 2026-03-30
>
> Dear Reviewer XnV3, we sincerey appreciate your review, recognition & interest. `LINK`: https://anonymous.4open.science/r/Swordsman-ICML/rebuttal.md
>
> >**Q1. More Related Works**
>
> Thanks for your suggestion! We promise to include these works in final version: ①Extensions of dLLMs to the multimodality: LLaDA-V, MMaDA, LLaDA-o, Omni-Diffusion, Lumina-DiMOO, LaViDa, Dimple & MMaDA-Parallel. ②Works on dLLMs decoding optimization: SlowFast, Learn2PD, Fast-dLLM v2, Dynamic-dLLM, ES-dLLM & Elastic-Cache. ③Others: LongLLaDA for generating extremely long texts & d1 for enhancing reasoning
>
> >**Q2. Expression Improvement**
>
> Thanks for constructive input! We acknowledge our expression needs further improvement. We will include following explanation: Syntactic constituents: *such as verb phrases and noun phrases*. Semantic constituents: *such as semantically tightly coupled phrase segments, e.g., ORANGE JUICE*. Two concepts differ but also overlap. As for exposition, we plan to restructure is as: Word importance is generally associated with prediction difficulty→ERH→Change thinking: consider future uncertainty reduction→constituent boundaries exhibit high uncertainty→greater potential for uncertainty reduction→analyse entropy to partition boundaries (blocks). We welcome your further insights, and it will be reflected in final version
>
> >**Q3. Entropy Visualization & Expansion**
>
> Dream's entropy visualization is shown as `Fig 13`. We briefly conduct our strategy on MMaDA but obtain some rough results difficult to present clearly, finding very limited or even no performance gain. We suspect this is related to tasks. Image understanding's text lengths are relatively short, particularly in binary classification, which offer limited gains. Similarly, in image generation, there is no semantic block concept in the image, rendering the strategy ineffective. However, we believe this is a promising direction, requiring more specialized designs or strategies. We even believe that entropy-guided remasking could be more effective than only focusing on unmasking. Limited by time and resource, we regret that we were unable to conduct further exploration with more models or designs; we sincerely hope this will not affect the rating. Thanks
>
> >**Q4. Evaluation on Long-context & Hallucination**
>
> Thanks for your suggestion. We add evaluation on long texts in `Fig 14` As hallucination remains under-explored in this field, we simply follow TraceDet [1] for DLM-tailored detection in `Tab 11`
>
> [1] TraceDet: Hallucination Detection from the Decoding Trace of Diffusion Large Language Models
>
> >**Q5. More Visualization**
>
> Entropy visualization across different layers and steps is shown in `Fig 15`. For code, we have consulted AC via a confidential comment for approval. Once granted, relevant scripts will be uploaded to anonymous repository immediately. Even if not approved now, we commit to open-sourcing it immediately upon acceptance to facilitate your further exploration. Thanks for your interest and attention
>
> > **Q6. Comparison on Dynamic Parallel Unmask (DPU) & SlowFast**
>
> Thanks for the reference. First, both SlowFast and DPU are fundamentally parallel decodings, as any masked tokens meeting their specific unmasking threshold will be unmasked together. The key difference is whether unmasking threshold is dynamic or static. DPU's threshold is typically composed of criteria such as confidence or entropy, which updates dynamically as decoding progresses. In contrast, SlowFast employs a static unmasking threshold. Additionally, a key feature of SlowFast is its another static threshold, which is used not for unmasking but for narrowing its unmask scope. In short, DPU is a dynamic decoding within a fixed scope, SlowFast is a static decoding within a varying scope. Theoretically, they could be combined, but inapplicable to our method. While we can just replace our DPU with SlowFast as shown in `Tab 12`
>
> >**Q7. Additional Concerns**
>
> ①We acknowledge that hyperparameter setting is heuristic. Regretfully, it is a common limitation faced by similar works and is also unavoidable for a training-free method. Without real-time prediction feedback to compute gradients for partitioning optimal block boundaries, we must rely on manually designed criteria to indirectly partition. ②Similar to many works originate from heuristic observations, entropy smoothing is our core insight. ③Under your instructions, we provide additional failure cases in `Fig 16`. ④Fig 1 is plotted based on the final layer output and its vertical axis discribes decoding steps. Although we respectfully believe that some issues are inherently difficult to resolve, we commit to refining them as much as possible in final version. Thanks
>
> ---
> Once again, thanks for your invaluable guidance. Having addressed all your concerns through the provided analysis and experiments, We hope these will lead to a positive reconsideration of our submission and welcome any further discussion. MANY THANKS!

---

> > ### Author Rebuttal · Reviewer_xViQ · 2026-04-01
> >
> > Thank you for the detailed response. I like this work very much and have raised my score to 5.

---

> > > ### Author Response · Authors · 2026-04-06
> > >
> > > Dear Reviewer xViQ, we would like to extend our deepest gratitude for your thorough review, insightful questions, and such generous support! We are truly honored that you like our work. Most importantly, we would like to express our special gratitude for your constructive suggestions. Your expert suggestions, such as the inclusion of more related works, the explicit notion of key concepts, the enhanced motivation exposition from ERH to entropy-driven block partitioning, and the supplement of further visualizations and analyses, have genuinely and significantly enhanced the quality, completeness, and interpretability of our work. We sincerely thank you once again for your effort.
> > >
> > > Regarding the code for entropy visualization, due to ICML's policy, we may not be able to give you it immediately. Therefore, we reiterate our firm commitment that the relevant code will be immediately open-sourced upon the acceptance of this submission. Thank you for your patience and interest.
> > >
> > > We genuinely hope to receive your continued support in the subsequent stages, which will be crucial to the success of this submission. MANY THANKS!

---

### Official Review · Reviewer_9Hph · 2026-03-08

**Soundness:** 3
**Presentation:** 3
**Significance:** 3
**Originality:** 3
**Overall Recommendation:** 4
**Confidence:** 4

**Summary:**

The paper proposes a training-free inference framework, Swordsman, that improves block-wise decoding by adaptively partitioning tokens into blocks based on semantic boundaries using entropy shifts between adjacent token predictions. Swordsman also adjusts unmasking thresholds based on block-averaged entropies. By using adaptive-length block partitions, Swordsman improves benchmark performance.

**Compliance With Llm Reviewing Policy:**

Affirmed.

**Final Justification:**

The authors addressed most of my concerns surrounding unclear presentation, questionable theoretical assumptions, comparison w/ related approaches, and variance measures in their experimental results.

There are still unaddressed weaknesses, but the strengths of the paper post-rebuttal outweigh the limitations (the variance measures confirm that their method is comparable in speed to baseline method Fast-dLLM, the authors provide a heuristic/abstract quantitative alignment metric for analyzing their method).

**Key Questions For Authors:**

- Swordsman is significantly worse on MBPP than AdaBlock in Table 1: would it be possible to evaluate with AdaBlock using the Fast-dLLM evaluation protocol?
- From Fig 2, it appears that tokens are partitioned based on sentence boundaries in practice (then one could partition tokens based on sentence boundaries rather than relying on entropy estimates). Are there cases where sentences consist of multiple blocks with the proposed method?
- Can the authors provide empirical comparison between their proposed sampler and the entropy-bounded sampler? [1]
- Can the authors provide quantitative results on the alignment with the proposed entropy-driven block partitioning and the constituency-parse structure, to supplement the empirical example in Fig 2? In particular, it would be insightful to understand whether "alignment" improves with the proposed entropy-driven block partitioning relative to fixed block partitioning [2]
- Could the authors provide experimental details for the constituency parser used in Fig 2?
- Could an off-the-shelf parser be used for determining block boundaries, and if so, can the authors provide discussion on how this would compare to entropy-based partitioning?

[1] Ben-Hamu et al. Accelerated Sampling from Masked Diffusion Models via Entropy Bounded Unmasking. NeurIPS 2025

[2] Arriola et al. Block Diffusion: Interpolating Between Autoregressive and Diffusion Language Models. ICLR 2025

**Limitations:**

Yes

**Strengths And Weaknesses:**

Strengths
- **Thorough and impactful empirical analysis of adaptive block partitioning.** The authors justify their approach by showing the entropy evolution during decoding (Fig 1) and the empirical token separations from adaptive block partitioning (Fig 2).
- **Insightful theoretical justification for why entropy shifts can detect constituent boundaries.**
- **Novel decoding strategy that adjusts confidence thresholds using entropy-based difficulty estimates and decoding progress.** The proposed sampler is novel, intuitive and effective, yielding meaningful improvements on benchmark performance.
- **Proposed adaptive block partitioning yields meaningful benchmark improvements compared to fixed block partitioning.** It appears that Swordsman yields a better speed-quality tradeoff compared to prior approaches (Fig 4, however this figure is extremely difficult to parse, see weaknesses)

Weaknesses:
- **Questionable theoretical assumption in the entropy analysis.** The derivation relating entropy to candidate vocabulary size assumes a uniform distribution over the candidate token set (Line 215). However, token distributions are typically non-uniform and concentrated on a small number of tokens. It is unclear whether the theoretical argument still holds under realistic token distributions
- **Missing key experimental details.** The authors report throughput improvements under a dual cache configuration, but they do not provide a citation or definition for dual caching or provide justification for this improvement. The authors are missing key details on recording decoding speed, including: variance measures (important as the margin between fast-dLLM [2] and their method in Table 2 is small), batch size, and number of samples used. Also, it appears that tables include a mix of results reported in the experiments and from other papers (Line 204 right), but this is not explicitly mentioned in table captions
- **Core experimental analyses are poorly presented.** A core result is showing that the proposed decoding strategy yields a better speed-quality tradeoff relative to prior approaches (Fig 4). However, Fig 4 is extremely difficult to parse and the caption is very short. (In the bar charts, what is the difference between the bars and the markers? In the line plot, why does the accuracy for fast-dLLM [1] fluctuate between high and low accuracy as throughput increases? Some markers refer to different models, how is model selection labeled for markers without explicit text labels?)
- **Missing ablation on sampler components.** The proposed sampling strategy appears to have three major interacting components: 1) adaptive block partitioning, 2) using entropy to adjust confidence thresholds, and 3) calibrating entropy estimates with historical entropy from preceding blocks. However, these components are not ablated
- **Lack of comparison with entropy-bounded sampler [1], which also uses entropy estimates for parallel decoding.**

All-in-all I believe the paper has strong methodological advancements, but some aspects of the evaluation and presentation appear incomplete. I would be happy to raise my score if the stated weaknesses are sufficiently resolved.

[1] Ben-Hamu et al. Accelerated Sampling from Masked Diffusion Models via Entropy Bounded Unmasking. NeurIPS 2025

[2] Wu et al. Fast-dLLM: Training-free Acceleration of Diffusion LLM by Enabling KV Cache and Parallel Decoding. ICLR 2026

---

> ### Author Rebuttal · Authors · 2026-03-30
>
> Dear Reviewer 9Hph, We sincerely appreciate your review and feedback! We value each of them and reply as below. `LINK`: https://anonymous.4open.science/r/Swordsman-ICML/rebuttal.md
>
> >**W1. Non-Uniform Token Distribution**
>
> We fully agree with your point that in reality, token distributions are typically non-uniform. Therefore, we propose the *effective token search space* (line 214) to address this, corresponding to the *candidate vocabulary* we mentioned, where the distribution is approximately uniform. Thus, entropy is calculated based on the effective candidate vocabulary. Furthermore, we introduced a correction factor $\epsilon_{flat}$ to compensate for the deviation between the actual distribution and a uniform distribution, better aligning with the non-uniform nature. Thanks
>
> >**W2. More Experiment Details**
>
> Dual cache was proposed by Fast-dLLM, which provides a detailed definition and throughput improvements analysis. Currently, dual cache has become a standard setting for comparisons among methods. Following your guidance, `Tab 6` presents metrics and setup details for decoding speed. While Fast-dLLM shows comparable average performance to ours, its variance is higher, indicating less stable generation. Regarding the mixed reported results, due to character limits, we kindly refer you to Reviewer XnV3 W1&Q1 Part II. Thanks
>
> >**W3. Better Presentation**
>
> Thanks for your critique. Under your instructions, we redraw  is as `Fig 12`. Both accuracy and throughput are treated as dependent variables that vary based on the cache strategy; we apologize for any misunderstanding that implied a direct correlation between the two. Thanks
>
> >**W4. More Ablation**
>
> Thank you for pointing out this shortcoming. ①We integrate partial results from Table 1 and Tab 3 into `Tab 7.1` for a clearer adaptive block partitioning ablation. ②We convert Fig 5 into `Tab 7.2` to provide a clearer dynamic threshold ablation. ③We add a calibration ablation by historical entropy in `Tab 7.3`
>
> >**W5&Q3. Comparison with EB-Sampler**
>
> Thanks for offering the reference. Although both ours and the EB-sampler are based on MDM, they differ significantly. ①**Core Objective**: EB-Sampler focuses on which tokens can be decoded together in current step, representing a local parallel unmasking optimization. Ours focuses on how to partition sequence into blocks align with constituent boundaries, achieving a global block-wise scheduling and decoding strategy. ②**Operation Level**: EB-Sampler focuses on the max parallel unmasking token number within one step, operating on tokens within a block. Ours focuses on how to better partition blocks and further optimise unmasking within each block, operating on both blocks and tokens. ③**Entropy Usage Strategy**: EB-Sampler introduces Bounded Entropy to quantify joint dependence error for determining token number. Ours uses entropy shift to partition and average entropy to control unmasking. Overall, they are different types. Under your guidance, we replace dynamic threshold with EB in `Tab 8`
>
> >**Q1.Performance Gap with Adablock**
>
> That's insightful. The performance gap mainly because cited AdaBlock results involve undisclosed post-processing. We include a detailed ablation in `Tab 9`, where the method without post-processing follows Fast-dLLM's evaluation protocol
>
> >**Q2&Q6. More Partition Feasibility and Multiple Blocks**
>
> There may be a misunderstanding. During inference, sequence is progressively produced by unmasking masked tokens. It is unable to partition blocks directly using sentence boundaries or parser analysis. Our entropy-based block partitioning may split one sentence into multiple blocks, as shown in Fig 2: one color segment (sentence) contains multiple cylinders (blocks)
>
> >**Q4. Quantitative & Visualized Alignment**
>
> Great suggestion! We designed a metric to evaluate the alignment degree between blocks and sentences. `Tab 10` compares the alignment metric of ours against fixed partitioning methods. Furthermore, `Fig 11` visualizes the alignment between our partitioned blocks and constituency boundaries, and white dashed lines in Figure 1 also demonstrate this alignment. Thanks again for your valuable input!
>
> >**Constituency Parser Details**
>
> Our constituency parser draws on《Syntactic Structures》and《Speech and Language Processing》. Briefly, it splits sentence into words and performs POS tagging. Based on predefined context-free grammar, it performs top-down parsing with backtracking to construct a hierarchical syntactic tree. We have sought AC’s advice via confidential comment. Once approved, its scripts will be uploaded to repository
>
> ---
> Once again, we sincerely thank you for your comment and advice. We commit to incorporating all above additional content into final version, truly honoring your time and effort. As all other ratings are positive, we eagerly look forward to your further support, which is crucial to our work. We also welcome further discussion. MANY THANKS!

---

> > ### Author Rebuttal · Reviewer_9Hph · 2026-04-02
> >
> > Thank you for thoroughly addressing my concerns! I have one follow-up question (I have raised my score to 4 as most of my questions/concerns have been thoroughly addressed):
> >
> > > W2: Speed variance measures for comparing Fast-dLLM and Swordsman
> >
> > Thank you for the new variance analyses (Table 6). However, it confirms that the throughput of Swordsman is within standard deviation of Fast-dLLM (both methods appear to be comparable with respect to decoding throughput), and I do not observe the variance of Fast-dLLM's throughput being higher (can the authors provide inline reported numbers to verify this claim?)
> >
> > > Q4: Quantitative Alignment
> >
> > Can the authors clarify how the quantitative alignment was computed? What is the mathematical formulation of the metric?
> >
> > > Constituency Parser Details
> >
> > Can the authors provide a reference/citation for the predefined context-free grammar used? My main concern was about the lack of detail in describing the experimental setup of the constituency parser in the paper contents, rather than explicitly providing the code implementation.

---

> > > ### Author Response · Authors · 2026-04-06
> > >
> > > Dear Reviewer 9Hph, we deeply appreciate your response and critical continued support. We are thrilled to discuss your follow-up question in more detail
> > >
> > > >**Speed Variance Measures**
> > >
> > > Thank you for your meticulous observation. Guided by your suggestion, we add variance metrics in `Tab 15` (https://anonymous.4open.science/r/Swordsman-ICML/rebuttal.md) to complement the standard deviation in `Tab 6`. As shown, Swordsman maintains lower variance than Fast-dLLM across various cache strategies and models. Furthermore, we also visualize scatter plots in `Fig 18`. Swordsman not only achieves a higher mean TP but also yields a more concentrated and stable distribution. In contrast, Fast-dLLM shows greater overall fluctuation, with several points severely deviating from the mean. Overall, Swordsman is not only superior in speed but also more stable and robust
> > >
> > > We also honestly admit that although ours enhances TP, it does not achieve a massive leap over Fast-dLLM. We suspect that we approach the upper limit for TP gains within block-wise methods. Similar works (GeoBlock/WINO/AdaBlock) also show limited improvements. Achieving the next leap will likely explore new perspectives. However, compared to vanilla LLaDA, TP increase from 3.32 to 44.89 is atisfactory. We appreciate this insightful question, which drives us to think more deeply about our method boundary. We will incorporate this discussion into a limitation in final version
> > >
> > > >**Quantitative Alignment**
> > >
> > > Limited by characters, we cannot provide a detailed explanation. We sincerely appreciate the opportunity to further clarify. We quantify the alignment by calculating the ratio of precisely aligned blocks to the total blocks. A clearer and more detailed mathematical formulation is as follows:
> > >
> > > For a sentence with $L$ tokens, we denote its token index set as $\mathcal{T} = \lbrace 1, 2, \dots, L \rbrace$. A contiguous block interval on $\mathcal{T}$ is defined as $[s, e] = \lbrace x \in \mathcal{T} \mid s \le x \le e\rbrace$, where $s$ and $e$ denote start and end indices. Therefore, Swordsman partitions $\mathcal{T}$ into $m$ disjoint contiguous blocks, denoted as partitioned block set (Blue in Fig 11) $\mathcal{B} = \lbrace B_1, B_2, \dots, B_m\rbrace$, where each $B_i = [s_i, e_i]$ and $\bigcup_{i=1}^{m} B_i = \mathcal{T}$. Similarly, syntactic block set (Orange in Fig 11) $\mathcal{P} = \lbrace P_1, P_2, \dots, P_n\rbrace$ obtained from a constituency parser also partitions $\mathcal{T}$ into $n$ disjoint contiguous intervals. If a syntactic block $P_j$ is entirely contained within a partitioned block $B_i$ (i.e., $\exists B_i\in\mathcal{B}\text{ s.t. }P_j\subseteq B_i$), it is a precise alignment; otherwise, it is imprecise. Accordingly, alignment degree ($AD$) is quantified by calculating ratio of precise alignments to total number of alignments:
> > >
> > > $$AD=\frac{1}{n}\sum_{j=1}^{n}\mathbb{I}(\exists B_i\in\mathcal{B}\text{ s.t. }P_j\subseteq B_i)$$
> > > Higher $AD$ indicates greater alignment degree between partitioned and syntactic blocks. We appreciate your insightful question and the chance to elaborate on this metric. We ensure adding these detailed explanations to final version
> > >
> > > >**Constituency Parser Details**
> > >
> > > Thanks for your careful attention to our setup details, which helps us make our work more solid. The references for the predefined context-free grammar (CFG) are [1-3]. Additionally, we will include the following details of constituency parser in final version:
> > >
> > > We adopt a CFG constituency parser following Penn Treebank conventions. Non-terminals include standard types (`S`, `NP`, `VP`, `PP`, `SBAR`, `SINV`, `SQ`, `ADJP`), while pre-terminals use standard POS tags (`PRP`, `DT`, `NN`, `VBD`, `IN`, `CC`). Representative productions include: `S → NP VP`, `NP → PRP | DT NN | NP PP | NP CC NP`, `VP → VBD NP | VBD PP | VP CC VP`, `PP → IN NP`
> > >
> > > Parsing begins by tokenizing and POS-tagging the input to establish leaf and pre-terminal layers. The parser then performs top-down recursive expansion from S, requiring each applied production to cover an exact contiguous span of the input. Backtracking is triggered whenever a partial expansion cannot be extended to cover the full sentence, and parsing succeeds when a complete derivation with no remaining tokens is found
> > >
> > > [1] Chomsky, Noam. Syntactic structures. Walter de Gruyter, 2002
> > >
> > > [2] Jurafsky, Daniel, and James H. Martin. "Speech and Language Processing: An Introduction to Natural Language Processing, Computational Linguistics, and Speech Recognition."
> > >
> > > [3] Bies, Ann, et al. "Bracketing guidelines for Treebank II style Penn Treebank project." University of Pennsylvania 97, 1995
> > >
> > > ---
> > > Once again, we extend our deepest gratitude for your precious time and professional advice, which truly elevates the quality, completeness and interpretability of our work. Most importantly, we deeply appreciate your generous support. We genuinely hope to receive your continued further support in subsequent stages. MANY THANKS!

---

### Official Review · Reviewer_XnV3 · 2026-03-12

**Soundness:** 2
**Presentation:** 3
**Significance:** 3
**Originality:** 2
**Overall Recommendation:** 4
**Confidence:** 3

**Summary:**

This paper proposes Swordsman, a training-free block-wise decoding framework for Diffusion Language Models (DLMs). To mitigate the semantic fragmentation caused by fixed-length chunking, the authors introduce an entropy-shift detection mechanism to adaptively place block boundaries at local maxima of token uncertainty, coupled with a dynamic threshold unmasking strategy that balances parallelism and reliability by monitoring intra-block entropy decay and inter-block difficulty. Without additional training, Swordsman achieves favorable accuracy-speed tradeoffs across multiple backbones and benchmarks, notably outperforming Fast-dLLM in code and mathematical reasoning tasks while maintaining comparable throughput.

However, three weaknesses persist: (1) insufficient differentiation from AdaBlock regarding the signal selection and fair comparison protocols for semantic-aware adaptive blocking; (2) lack of empirical validation for the approximations assumed in Section 3.2's theoretical analysis; and (3) unquantified overhead from the per-block entropy refresh forward passes. Overall, the work presents a competitive contribution to efficient DLM inference.

**Compliance With Llm Reviewing Policy:**

Affirmed.

**Key Questions For Authors:**

1. **Comparison to AdaBlock:**
   Could the authors more clearly articulate the conceptual difference between Swordsman and AdaBlock, and, if possible, provide a more directly aligned comparison under a unified evaluation protocol?

2. **Theory validation:**
   The intuition in Sec. 3.2 is reasonable, but it relies on several simplifying assumptions. Could the authors provide additional empirical evidence showing when the entropy-based approximation is reliable in practice?

3. **Overhead analysis:**
   Since Swordsman refreshes entropy after each decoded block, could the authors quantify the extra computation introduced by this step, for example through a wall-clock or FLOPs breakdown?

**Limitations:**

While the paper presents a practical and promising training-free decoding strategy, the current version would be further strengthened by a clearer positioning against the most closely related adaptive block decoding methods, more explicit validation of the theoretical assumptions in Sec. 3.2, and a more transparent breakdown of the additional computation introduced by entropy refresh. These issues do not undermine the overall value of the work, but addressing them would make the claims more convincing and improve the paper’s calibration for a top-tier venue.

**Strengths And Weaknesses:**

### Strengths

- The method is training-free and plug-and-play, requiring no modification of model parameters or additional training. It adjusts the decoding process via entropy-driven adaptive partitioning and dynamic thresholding, compatible with KV caching, demonstrating strong engineering practicality and deployment flexibility.
- The experimental design is comprehensive, covering multiple backbones, benchmarks, and cache configurations. The results exhibit consistency across models and tasks, providing substantial empirical support for the core claims.
- On critical tasks such as code generation and mathematical reasoning, the method achieves considerable accuracy improvements while maintaining comparable or superior inference speed, exhibiting a competitive accuracy-speed tradeoff.


### Weaknesses

- The differentiation from AdaBlock remains insufficient. Both are training-free, semantic-aware adaptive blocking methods, yet employ distinct boundary signals (entropy shift versus confidence dynamics). The manuscript fails to clarify their theoretical equivalence, complementarity, or respective applicable conditions. Moreover, reliance on cited values rather than direct alignment under unified protocols obscures whether observed gains stem from methodological novelty or comparative protocol discrepancies.
- The additional forward-pass overhead required for inter-block entropy refresh is not individually quantified. Although overall throughput remains competitive, the absence of isolated cost breakdowns for refresh operations precludes clear assessment of whether benefits consistently outweigh costs as block counts, sequence lengths, or task types vary.

---

> ### Author Rebuttal · Authors · 2026-03-30
>
> Dear Reviewer XnV3, we sincerey appreciate your careful review and constructive comments. We value each of them and provide responses below. `LINK`: https://anonymous.4open.science/r/Swordsman-ICML/rebuttal.md
>
>  > **W1&Q1. Difference from AdaBlock**
>
> **Difference between Ours & Adablock**
>
> Both AdaBlock and Swordsman are inspired by the observation that fixed-length block partitioning may fragment semantic constituents, and both attempt to address this through adaptive blocking strategies. However, the two approaches differ significantly in theoretical foundations and practical contributions, as we clarify across three dimensions below.
>
> ① **Theoretical Foundations**: AdaBlock justifies its adaptive partitioning primarily on empirical statistical analysis of model decoding behavior, observing potential semantic phase phenomena through confidence variation patterns. Swordsman, on the other hand, further incorporates an information-theoretic perspective to provide more formal theoretical support for the connection between entropy shifts and semantic phrase chunk boundaries. That is, rather than constructing heuristic strategies purely from empirical observations, we provide a more explicit theoretical explanation (`Sec 3.2`) for "why entropy shifts can serve as boundary signals."
>
> ② **Block Partitioning Criteria and Granularity**: AdaBlock is essentially a heuristic block partitioning method that strongly relies on predefined boundary tokens, where the block endpoint is determined by the confidence of preset end-token candidates within a local window. As a result, the partitioning outcome is heavily constrained by hand-crafted rules, yielding coarser-grained segmentation closer to the sentence level. Swordsman does not rely on any predefined boundaries. Instead, it naturally characterizes constituents through entropy shifts between adjacent tokens, allowing block boundaries to be directly determined by the evolution of uncertainty within the sequence, thereby achieving more natural and fine-grained semantic segmentation.
>
> ③ **Contribution Scope**: AdaBlock focuses exclusively on inter-block scheduling, i.e., determining where a block ends, while leaving the intra-block decoding process largely unchanged. Swordsman addresses both levels: it proposes entropy-driven adaptive block partitioning and introduces a dynamic-threshold parallel unmasking mechanism that adapts decoding confidence to block-specific difficulty. Our framework thus provides a more complete solution to the quality-speed trade-off in DLM inference.
>
> **Re-evaluate Adablock**
>
> As AdaBlock had not yet open-sourced its code at the time of the paper submission deadline, we could only cite the corresponding results from its paper as a comparison, also noted in Appendix A along with highlighting certain metric inconsistencies. As AdaBlock has now been publicly accessible, we re-evaluated it under the same evaluation protocol of Fast-dLLM, with results shown in `Tab 5`. We commit to updating the experimental results in the camera-ready version.
>
>  > **W2&Q3. Cost of Entropy Mechanism**
>
> Thank you for raising this important efficiency concern. Please refer to our response to W1 of Reviewer EopJ.
>
>  > **Q2. Additional Empirical Evidence of the Entropy-based Approximation**
>
> Thanks for your question. Fig 2 in our paper visually demonstrates that the blocks (yellow bars) partitioned based on entropy shift (black arrows) exhibit good alignment with the actual constituents derived from constituency parsing (the left yellow parse tree). To offer more quantitative support, we conducted a statistical analysis of case studies on entropy smoothing and shifts. Please refer to `Fig 11`.
>
> ---
> Once again, we sincerely thank you for your time and positive feedback. We are fully committed to incorporating your recommended analyses and experiments to further strengthen the final version. We hope our rebuttal has successfully addressed your constructive concerns and look forward to your continued support for our submission. We welcome any further discussion during the discussion period and would be more than happy to address any additional questions you may have. Thank you!

---

> > ### Author Rebuttal · Reviewer_XnV3 · 2026-04-07
> >
> > Thank authors for the effort on their rebuttal. My question is partially resolved. I decide to keep my score.

---

> > > ### Author Response · Authors · 2026-04-08
> > >
> > > Dear Reviewer XnV3, we sincerely thank you for your feedback and continued support. We are gratified to have addressed some of your concerns. We highly value your comments. Regarding the remaining concerns, we suspect they may involve more macroscopic and abstract aspects that are difficult to resolve within a short rebuttal. Nevertheless, we still sincerely hope to ensure your satisfaction. If it is convenient and you are willing, we welcome your further questions, and we are more than happy to do our best to address your concerns. If there are indeed elements that require macroscopic revisions, we commit to working hard on these and ensure they are reflected in the final version. Thank you once again for your time, effort, suggestions, and support! MANY THANKS!

---

### Decision · Program_Chairs · 2026-04-30

**Decision:**

Accept (regular)

**Comment:**

This paper addresses the *semantic fragmentation* in Diffusion Language Models (DLMs) using a training-free framework named Swordsman, which consists of entropy-driven adaptive partitioning and dynamic parallel unmasking. The experiments demonstrate state-of-the-art performance on benchmarks like code generation and math reasoning while being plug-and-play for existing DLM architectures.

The reviewers initially raised concerns regarding the differentiation from prior work and computational overhead, which were addressed during the rebuttal:
- Novelty & Motivation: Reviewers recognized the motivation as intuitive and novel, noting that the alignment with linguistic constituents is a meaningful advancement.
- AdaBlock Comparison: Authors clarified that Swordsman uses information-theoretic signals (entropy) for fine-grained partitioning, whereas AdaBlock relies on heuristic end-token candidates for coarser segmentation.
- Computational Cost: The rebuttal provided a FLOPs analysis showing the entropy mechanism adds only ~0.01% overhead per forward pass, which reviewers found convincing.
- Empirical Robustness: Additional variance metrics and alignment degree (AD) scores were provided to prove that Swordsman is more stable and better aligned with syntactic structures than fixed-block baselines.

In summary, the submission presents a technically sound and well-motivated approach to accelerating DLM inference. With final scores of Accept (5), Weak Accept (4), and Weak Accept (4), the consensus reflects that the strengths (particularly its training-free nature and efficiency gains) outweigh the minor remaining limitations.